# Human miRNA *miR-675* inhibits *DUX4* expression and may be exploited as a potential treatment for Facioscapulohumeral muscular dystrophy

Nizar Y. Saad [1], Mustafa Al-Kharsan[1,2], Sara E. Garwick-Coppens[1], Gholamhossein Amini Chermahini[1], Madison A. Harper[1], Andrew Palo[1], Ryan L. Boudreau [3] & Scott Q. Harper [1,4✉]

Facioscapulohumeral muscular dystrophy (FSHD) is a potentially devastating myopathy caused by de-repression of the *DUX4* gene in skeletal muscles. Effective therapies will likely involve *DUX4* inhibition. RNA interference (RNAi) is one powerful approach to inhibit *DUX4*, and we previously described a RNAi gene therapy to achieve *DUX4* silencing in FSHD cells and mice using engineered microRNAs. Here we report a strategy to direct RNAi against *DUX4* using the natural microRNA *miR-675*, which is derived from the lncRNA *H19*. Human *miR-675* inhibits *DUX4* expression and associated outcomes in FSHD cell models. In addition, *miR-675* delivery using gene therapy protects muscles from DUX4-associated death in mice. Finally, we show that three known *miR-675*-upregulating small molecules inhibit *DUX4* and DUX4-activated FSHD biomarkers in FSHD patient-derived myotubes. To our knowledge, this is the first study demonstrating the use of small molecules to suppress a dominant disease gene using an RNAi mechanism.

[1] Center for Gene Therapy, the Abigail Wexner Research Institute at Nationwide Children's Hospital, Columbus, OH, USA. [2] Department of Neurology, University of New Mexico Health Sciences Center, Albuquerque, NM, USA. [3] Department of Internal Medicine, Fraternal Order of Eagles Diabetes Research Center, Abboud Cardiovascular Research Center, Carver College of Medicine, University of Iowa, Iowa City, IA, USA. [4] Department of Pediatrics, The Ohio State University College of Medicine, Columbus, OH, USA. ✉email: scott.harper@nationwidechildrens.org

Facioscapulohumeral dystrophy (FSHD) is among the most commonly inherited muscular dystrophies, estimated to affect as many as 870,000 individuals. Classical descriptions of FSHD presentation include progressive muscle weakness in the face, shoulder-girdle and arms, but presentation is often non-uniform[1–3]. For example, 25% develop lower limb or abdominal weakness leading to wheelchair dependence, while others remain ambulant or asymptomatic. Moreover, although FSHD does not typically impact lifespan, some patients may develop life-threatening respiratory muscle complications. Age-at-onset can range from early childhood to adulthood, with earlier onset is often associated with more severe disease[4].

FSHD is caused by aberrant expression of the *double homeobox 4 gene* (*DUX4*), which produces a transcription factor that is toxic to skeletal muscle[5–13]. *DUX4* is normally functional during the four-cell stage of human development but repressed thereafter in essentially all other tissues, except the testes, skin, and thymus[14–17]. In skeletal muscles of people with FSHD, genetic and epigenetic factors conspire to permit *DUX4* de-repression, which then initiates several aberrant gene expression cascades, including those involved in muscle differentiation and atrophy, oxidative stress, inflammation, and cell death[5–9,13,18–21].

There are currently no approved treatments that slow FSHD progression or improve muscle weakness. The most direct route to FSHD therapy will involve inhibiting *DUX4* in muscle. Gene silencing by RNA interference (RNAi) is one powerful approach to inhibit *DUX4*. RNAi is mediated by ~22 nucleotide (nt), non-coding microRNAs in complex with cellular gene silencing machinery, which together are called the RNA-induced silencing complex (RISC). The microRNA component of RISC base-pairs with target transcripts to trigger sequence-specific gene silencing. Importantly, artificial microRNAs can be engineered to trigger RNAi against disease genes. Historically, RNAi-based therapies have relied upon two major strategies to silence dominant disease genes: (i) delivery of siRNA oligonucleotide drugs to permissive target cells or tissues; or (ii) gene therapy in which designed microRNA or shRNA expression cassettes are delivered with viral vectors and expressed intracellularly. To date, three oligonucleotide-based RNAi drugs—Alnylam's Patisiran, Oxlumo, and Givlaari—are FDA approved, while several promising gene therapy strategies for dominant diseases are still emerging. Using either strategy for FSHD therapy requires delivery of an RNAi product to muscle. Currently, intact muscle is a challenging target for oligonucleotide delivery, but importantly, AAV gene therapy vectors can achieve widespread and long-term transduction of virtually the entire musculature following a single systemic injection. Thus, gene therapy is currently the best approach for delivering an inhibitory RNA to muscle. Nevertheless, the relative permanence of gene therapy could also be a disadvantage; for example, if unwanted side effects developed from inhibitory RNA over-expression, it is currently impossible to turn the vector off or down. Here we pursued an alternative approach to induce RNAi against a dominant disease gene that circumvented delivery problems inherent with oligo-based RNAi systems, and would potentially allow tunability or stoppage of treatment if deleterious effects emerged. Specifically, we hypothesized that the *DUX4* transcript may be naturally regulated by one or more of the 1917 known endogenous human microRNA genes, and that it may be possible to upregulate a *DUX4*-targeting microRNA with a small molecule, thereby offering a potential drug-based RNAi treatment for FSHD. Unfortunately, prior to this study, no endogenous microRNAs targeting *DUX4* had been identified. We, therefore, initiated this study with the goal of identifying endogenous human microRNAs capable of silencing *DUX4*. To do this, we used bioinformatics tools to predict prospective *DUX4*-targeting human miRNAs and then empirically

tested their ability to silence *DUX4*. Among numerous candidates tested, we identified *miR-675* as a *DUX4* regulator. Here, we provide evidence that *miR-675* directly targets *DUX4* mRNA, inhibits DUX4-responsive biomarkers, and counteracts DUX4-induced toxicity in skeletal muscle and non-muscle cells. In addition, using AAV-based gene therapy, we show that *miR-675* can inhibit DUX4-associated histopathology in an FSHD mouse model. Finally, we validated previous studies showing that *miR-675* could be increased by estrogen/progesterone and melatonin, and then used these small molecules to upregulate endogenous *miR-675* and decrease *DUX4* and DUX4-responsive biomarkers in human FSHD myotubes. To our knowledge, this is the first study demonstrating the use of small molecules to suppress a dominant disease gene using RNAi. This study supports translating *miR-675* gene and drug therapies as prospective treatments for FSHD, and justifies identifying additional microRNAs that could be exploited for FSHD therapy. Furthermore, this work establishes a potential playbook to develop RNAi-based small molecule therapies for other dominant diseases.

## Results

**MicroRNA screen identifies human *miR-675* as a DUX4 inhibitor**. To investigate the hypothesis that *DUX4* was regulated by endogenous miRNAs, we first used the miRNA target prediction algorithm miRBase to identify miRNAs potentially targeting *DUX4* 3′ end sequences[22]. We tested 23 predicted microRNAs using a dual-luciferase reporter assay, but unfortunately none caused strong *DUX4* silencing (details in Supplementary Note 1 and Supplementary Fig. 1). We next tested an alternative target prediction algorithm, called PITA (https://genie.weizmann.ac.il/pubs/mir07/mir07_prediction.html), which attributes higher rank scores to miRNAs with multiple putative binding sites on a target transcript[23]. We also expanded the sequence search to include the entire *DUX4* ORF and 3′UTR, as data emerged suggesting that miRNA regulation can also occur in coding regions[24]. *Hsa-miR-675* was among the top hits predicted by PITA, with 14 putative binding sites located throughout the *DUX4* ORF and 3'UTR, including several with potentially high affinity. Based on these data, we focused on *miR-675* as our top candidate from the PITA algorithm and switched our Luciferase assay screening strategy to include a construct containing all *DUX4* pre-mRNA sequences (ORF and 3′ UTR), including potentially retained introns (called DUX4-FL, where FL = full-length). We then tested the ability of *miR-675* to silence *DUX4* sequences by luciferase assay in co-transfected HEK293 cells. To deliver *miR-675* to cells, we initially used a U6 promoter-driven *miR-675* expression plasmid (U6.MIR675), as well as a construct expressing the *miR-675* precursor, the lncRNA *H19* (CMV.H19) (Fig. 1a). Importantly, U6.MIR675 and the *miR-675* precursor *H19* reduced DUX4-FL luciferase activity by $35 \pm 3\%$ and $36 \pm 2\%$, respectively ($P < 0.0001$, ANOVA, $N = 3$), while the miLacZ negative control had no impact (Fig. 1a). As an additional control, we found that U6.MIR675 had no impact on *Renilla* luciferase that lacked *DUX4* sequences, indicating that it does not target *Renilla* luciferase alone (RenLuc-backbone). The level of knockdown achieved by *miR-675* against a *DUX4*-attached luciferase target was the best we had so far observed from a natural microRNA, but still did not approach the >80% knockdown elicited by *miDUX4.405*, which contains 22 nt of perfect complementary with *DUX4* and was designed for optimal gene silencing[21,25–27] (Fig. 1b). Since the PITA algorithm predicted binding sites in both the *DUX4* ORF and 3′UTR regions, we also tested U6.MIR675 against the *Renilla* luciferase constructs containing only the *DUX4* ORF (DUX4 ORF) or *DUX4* 3′UTR (DUX4 3′UTR). U6.MIR675 significantly silenced each construct by

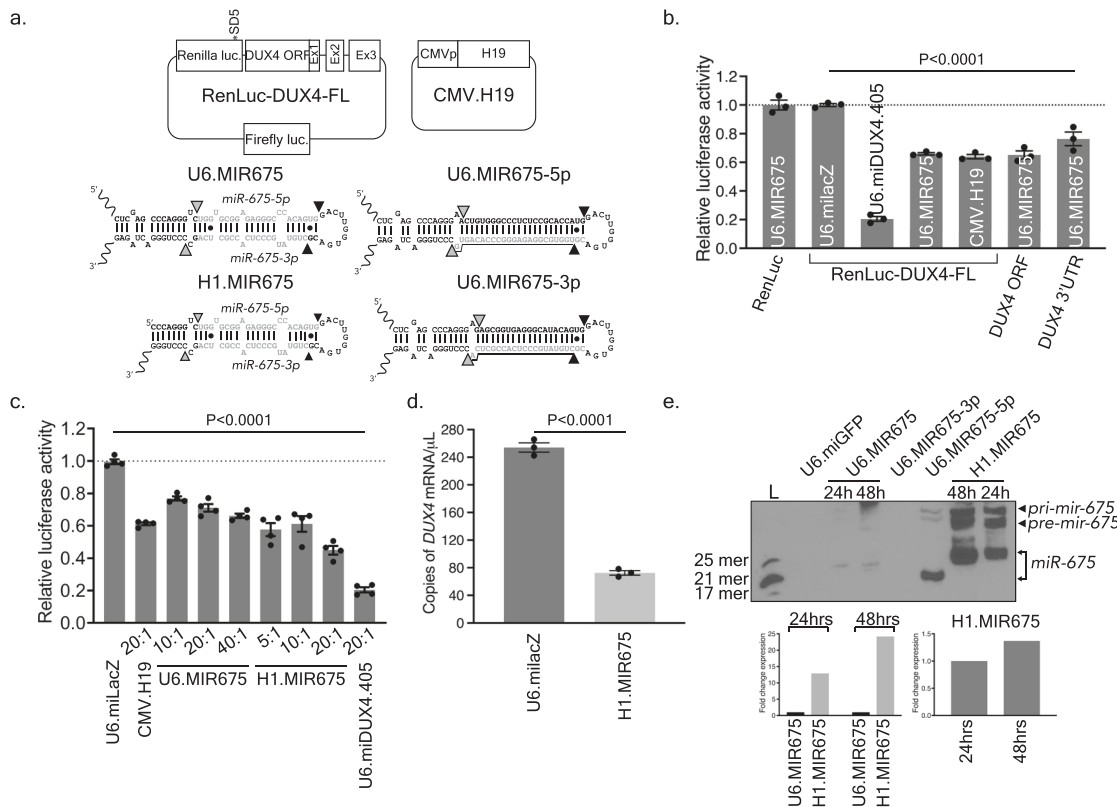

**Fig. 1 _miR-675_ targets the _DUX4 ORF_ and _3'UTR_. a** RenLuc-DUX4-FL dual-luciferase reporter and CMV.H19 constructs, and U6.MIR675, H1.MIR675, U6.MIR675-3p, and U6.MIR675-5p miRNAs. RenLuc-DUX4-FL contains _DUX4 ORF_ and _3'UTR_ sequences fused to _Renilla_ luciferase after the stop codon. Exons 1–3 are indicated (Ex1,2,3). *SD5 indicates silent mutation of a cryptic splice donor in _Renilla_ luciferase. Firefly luciferase is used as transfection control. **b** U6.MIR675 reduced relative _Renilla_ luciferase activity in constructs containing full-length _DUX4_, _DUX4 ORF_ only, or _DUX4 3' UTR_ only (35 ± 3%, 34 ± 2%, and 24 ± 5%, respectively; $P < 0.0001$). _H19_ reduced relative _Renilla_ luciferase activity from the RenLuc-DUX4-FL by 36 ± 2% ($P < 0.0001$). All readings were normalized to Firefly luciferase and then to the milacZ negative control. U6.MIR675 did not reduce relative _Renilla_ luciferase activity from the non-targeting RenLuc control plasmid (RenLuc). The U6.miDUX4.405 positive control is a published artificial microRNA with robust _DUX4_ silencing activity. Data indicate mean luciferase activity ± SEM ($N = 3$ independent experiments). **c** Relative _Renilla_ luciferase activity was reduced in a dose-dependent manner by two _miR-675_ constructs: U6.MIR675 and H1.MIR675. Molar ratios of miRNA to reporter plasmid are indicated and data normalized to the miLacZ negative control. U6.MIR675 significantly reduced relative _Renilla_ luciferase activity by 23 ± 3% (10:1 ratio), 29 ± 3% (20:1), and 34 ± 3% (40:1). H1.MIR675 performed better at lower doses, reducing relative _Renilla_ luciferase activity by 42 ± 4%, 39 ± 5%, and 55 ± 3%. All samples were significantly different from the miLacZ control ($P < 0.0001$). Data represent mean normalized _Renilla_ luciferase activity ± SEM ($N = 4$ independent experiments). **d** Droplet digital PCR (ddPCR). H1.MIR675 significantly reduced _DUX4_ mRNA levels by 71 ± 2% ($P < 0.0001$, ANOVA, $N = 3$) compared to the miLacZ negative control, 48 h following transfection of HEK293 cells with a 1:3 ratio of CMV.DUX4-FL/CMV.eGFP and H1.MIR675. Results represent the average absolute _DUX4_ mRNA concentration (copies/µL) ± 95% poisson confidence interval. Data represent mean relative concentration (copies/µL) ± SEM ($N = 3$ independent experiments). For **b**–**d**, One-way ANOVA followed by Dunnett's multiple comparison tests were performed for statistical analyses. **e** Northern blot using RNA extracted from HEK293 cells transfected with the indicated expression plasmids. U6.miGFP is a negative control. Source data are provided as a Source data file.

34 ± 2% and 24 ± 5%, respectively ($P < 0.0001$, ANOVA, $N = 3$) (Fig. 1b). These results supported prior evidence that natural miRNAs can initiate silencing of target genes through base pairing sequences in the _ORF_ as well as the _3'UTR_[24]. Finally, there are two species of _miR-675_ that originate from the _H19_ precursor, _miR-675-5p_ and _miR-675-3p_. Only _miR-675-5p_ was predicted to regulate _DUX4_ and we confirmed these results empirically (Supplementary Note 2 and 3 and Supplementary Fig. 2a, b). We also tested the functionality of our U6-driven expressing plasmids, by measuring the expression of their known target genes _SMAD1_, _SMAD5_, and _CDC6_[27] (Supplementary Fig. 2c).

To confirm the U6.MIR675 results, we tested a second _miR-675_ expression construct, utilizing the H1 promoter instead of U6. H1.MIR675 performed consistently better than the original U6.MIR675 construct, reducing relative _Renilla_ luciferase activity by 55 ± 3% (1.6 ± 0.2-fold better than U6.MIR675, $P < 0.0001$,

ANOVA, $N = 4$) (Fig. 1c). Using the CMV.DUX4-FL/CMV.eGFP, we also used droplet digital PCR (ddPCR) and QPCR to confirm that _miR-675_ significantly reduced _DUX4_ mRNA expression in co-transfected HEK293 cells (Supplementary Note 4, Supplementary Fig. 9 and Fig. 1d). To understand the difference in silencing produced by U6.MIR675 and H1.MIR675, we performed small transcript northern blots using RNAs isolated from HEK293s transfected with each construct. The H1.MIR675 construct produced 13-fold and 23-fold more full-length and mature _miR-675_ compared to U6.MIR675 at 24 h and 48 h post-transfection, respectively (Fig. 1e). In addition, the mature _miR-675_ species from both constructs were not identical in size, suggesting differential processing by microRNA biogenesis machinery. The low abundance of mature _miR-675-5p_ from U6.MIR675 could explain the low levels of inhibition when targeting the fully complementary target sequence (miR-675R). (Supplementary Fig. 2d). Based on these data, H1.MIR675

became our lead expression system for all subsequent experiments.

**miR-675-5p has multiple binding sites within the *DUX4* ORF and 3′UTR.** The PITA algorithm predicted 14 *miR-675* binding sites on *DUX4*[23]. We confirmed these predictions with a second platform, RNAHybrid, and then used these data to develop an in vitro binding assay to empirically identify functional *miR-675* target sites on *DUX4* (Supplementary Data 1 and Supplementary Fig. 3)[28]. To do this, we developed a molecular beacon-binding assay in which the *miR-675-5p* mature sequence was used as a DNA molecular beacon and the predicted target sites were incorporated individually in single stranded oligonucleotides (Fig. 2). Mature *miR-675* had significant binding affinity ($K_d$) with 8 of the 14 predicted *DUX4* target sites (Fig. 2). Two target sites were located in the 3′UTR (TS1340 and TS1471) and six were in the *DUX4* ORF (TS527, TS649, TS668, TS754, TS780, and TS1004). *miR-675-5p* showed highest binding affinity to TS780 in the *DUX4* ORF ($K_d = 6 \pm 1\,\mu M$) and TS1471 in the 3′UTR ($K_d = 3 \pm 1\,\mu M$) (Fig. 2b, c).

**miR-675 inhibits DUX4 mRNA and protein expression.** Luciferase reporter and in vitro binding assays suggested that *miR-675-5p* and the *H19* precursor regulated *DUX4* expression. In these experiments, *DUX4* sequences were present out of context, either attached as the 3′ UTR of *Renilla* luciferase or embedded within an oligonucleotide. We, therefore, sought to confirm *miR-675*-mediated silencing against a wild-type *DUX4* target and used western blotting as a secondary gene expression outcome measure. To do this, we co-transfected HEK293 cells, which do not normally express polyadenylated *DUX4*, with a DUX4 expression plasmid and various *miR-675* expression constructs. To express *DUX4*, we constructed a plasmid containing a CMV promoter driving full-length *DUX4* fused with a COOH-terminal V5 epitope, and included on the same plasmid a CMV-driven eGFP gene, which was used to normalize transfection efficiency (CMV.DUX4-FL/CMV.eGFP, aka DUX4/eGFP). We then performed a series of unblinded and blinded co-transfections of the DUX4/eGFP plasmid with various *miR-675* expression constructs, or controls, into HEK293 cells. Specifically, we transfected 3 different plasmids expressing *pri-mir-675* from different promoters (CMV, H1, and U6 promoters); two constructs engineered to preferentially express the −5p or −3p strands of *miR-675* (U6.MIR675-5p and U6.MIR675-3p); CMV.H19; the U6.miDUX4.405 artificial microRNA positive control; a U6.miLacZ negative control; and two oligonucleotide miR mimics, which were essentially siRNAs (*miR-675-5p* mimic and miR negative control mimic). We then collected protein extracts 48 h later and performed western blotting using antibodies to detect V5 epitope-tagged DUX4 protein (anti-V5) and eGFP. In total, we performed 6 independent transfections, protein collections, and western blot experiments on different days over the course of several weeks. The last four confirmatory experiments were performed using a blinded design, as described in the Methods section. If *miR-675* did indeed suppress *DUX4* expression, we were expecting mild to modest inhibition, as natural microRNAs typically contain sequence mismatches with target genes and do not direct gene silencing as efficiently as engineered microRNAs containing 22 nt miRNA:mRNA binding sites (e.g., *miDUX4.405*). Indeed, the *miDUX4.405* positive control reduced DUX4 protein by $93 \pm 4\%$ ($P < 0.0001$, ANOVA, $N = 6$), as we previously described[21,29] (Fig. 3a). Unlike *miDUX4.405*, *miR-675* did not contain perfect complementarity with *DUX4* at any predicted binding site (Fig. 2c), and so we were surprised to find that the H1.MIR675 expression plasmid produced robust *DUX4*

gene silencing of $91 \pm 4\%$ ($P < 0.0001$, ANOVA, $N = 6$), similar to that of *miDUX4.405* (Fig. 3a and Supplementary Fig. 4–8). All other *miR-675* constructs, including *H19* and the *miR-675-5p* mimic, reduced DUX4 protein expression but with variable and more modest efficiencies, ranging between 35 and 53%. As expected, the negative controls, U6.miLacZ and miR negative control mimic, did not inhibit DUX4 protein levels (Fig. 3a and Supplementary Fig. 4–8). Variability in expression and processing of primary *miR-675* transcripts into mature *miR-675-5p* sequences could account for the differences in silencing capability among the various *miR-675* expression constructs (Fig. 1e).

To confirm the western blot findings and test the specificity of *miR-675* repressive activity on *DUX4*, we introduced silent point mutations in the strongest *miR-675-5p* binding site on *DUX4*, target site 780 (TS780) (Fig. 2 and Supplementary Fig. 3b). These changes were designed to make the mutated TS780 sequence (TS780M) unrecognizable by *miR-675-5p*. Moreover, in the same construct, the *DUX4* 3′UTR was omitted, thereby eliminating the other validated high affinity binding site for *miR-675* (TS1471) as well as a second binding site with lower affinity (TS1340). We named this construct CMV.DUX4-miR-675Res (where miR-675Res = *miR-675* resistant). We cloned a similar construct in which the TS780 was deleted from our *DUX4* ORF luciferase reporter plasmid (RenLuc-DUX4 ORF-miR-675Res) (Fig. 3b). We used the same transfection strategy and outcome measures (western blotting or luciferase assay) described above but replaced the wild-type *DUX4* expression plasmids (i.e., DUX4/eGFP and RenLuc-DUX4-FL) with the *miR-675* resistant analogs (CMV.DUX4-miR-675Res/CMV.eGFP and RenLuc-DUX4 ORF-miR-675Res). Importantly, as measured by western blot and luciferase assay, none of the *miR-675* constructs were capable of suppressing DUX4 protein expression and relative *Renilla* luciferase activity from the *miR-675*-resistant mutants (Fig. 3b). Together, these results confirmed the specificity of *miR-675-5p*-mediated *DUX4* repression.

**miR-675 and its precursor H19 protected HEK293 from DUX4-induced cell death and inhibited DUX4 transactivation.** *DUX4* encodes a transcription factor that is toxic to numerous cells and tissues from multiple organisms, leading to cell death through apoptosis and possibly other mechanisms[6,7,9,30–35]. Indeed, HEK293 cells are killed by *DUX4* over-expression, and since *miR-675* can silence *DUX4* expression, we hypothesized that it could protect HEK293 cells from DUX4-induced cell death. To investigate this, we co-transfected HEK293 cells with H1.MIR675 and CMV.DUX4 expression plasmids, and measured cell viability over time using a Trypan blue cell viability assay (Fig. 4a). Cells co-transfected with *DUX4* and a negative control showed $23 \pm 2\%$ ($P < 0.0001$, ANOVA, $N = 3$) and $58 \pm 1\%$ ($P < 0.0001$, ANOVA, $N = 3$) cell death by 48 and 62 h (our longest time point), respectively, when compared to the mock-transfected cells (No *DUX4*). However, H1.MIR675 significantly protected HEK293 cells from DUX4-induced cell death up to 62 h after transfection as only $14 \pm 1\%$ of cells were dead ($P < 0.0001$, ANOVA, $N = 3$) similar to our positive control U6.miDUX4.405 that allowed the death of only $13 \pm 1\%$ of cells ($P < 0.0001$, ANOVA, $N = 3$).

Prior studies showed that DUX4 activates apoptosis in HEK293 cells, which can be measured by activation of Caspase-3/7[9,13]. Thus, to confirm the cell viability assay, we performed a second co-transfection experiment using Caspase-3/7 assay as an outcome measure for apoptosis. Cells co-transfected with *DUX4* and U6.miLacZ showed a $2.2 \pm 0.1$-fold increase ($P < 0.0001$, ANOVA, $N = 3$) in Caspase 3/7 activity compared to the Caspase-3/7 activity of mock-transfected (No *DUX4*) cells, indicating an increase in apoptosis. However, U6.miDUX4.405,

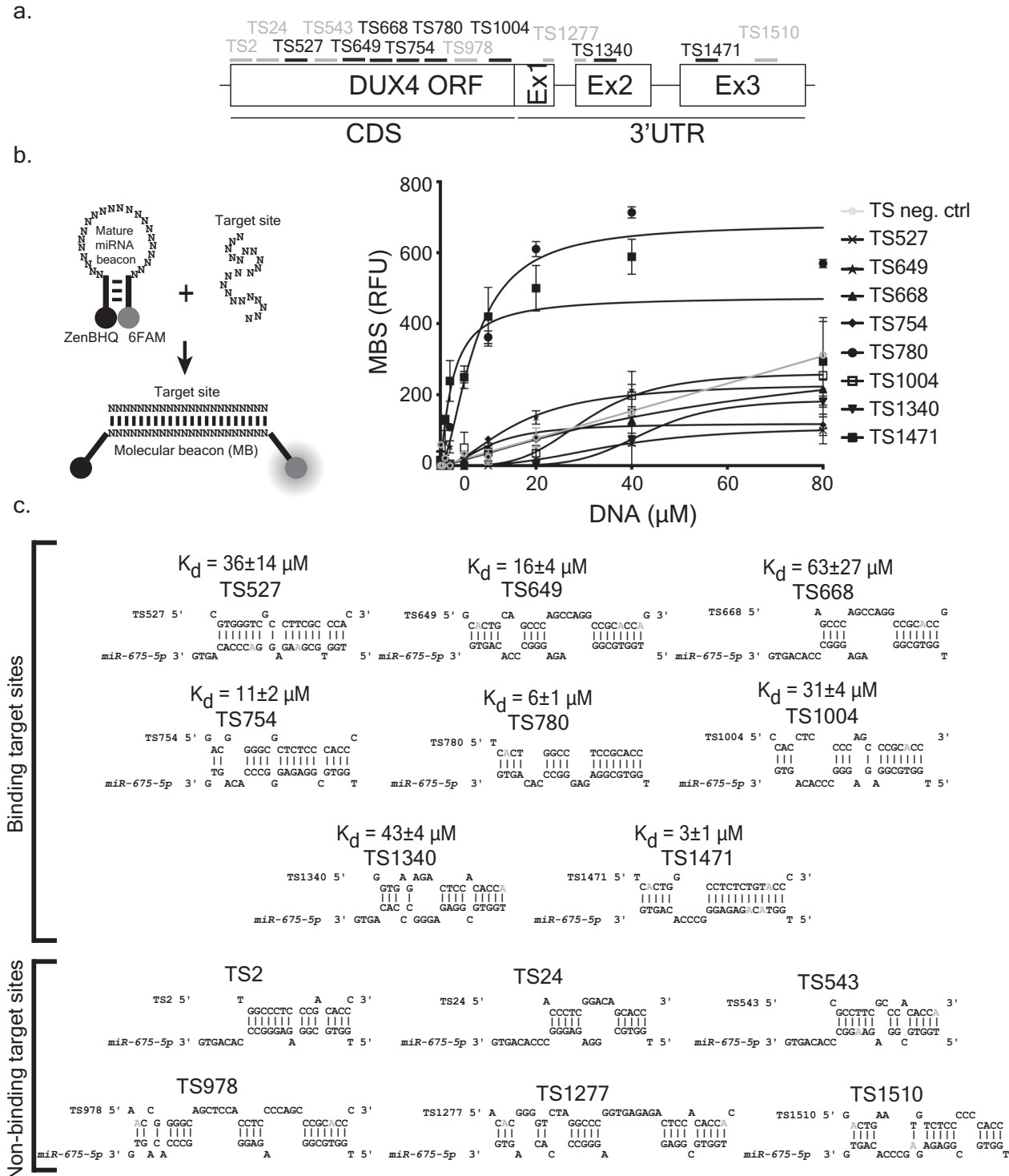

**Fig. 2 *miR-675-5p* targets multiple binding sites in the *DUX4 ORF* and *3'UTR*. a** Schematic of *DUX4* sequence showing predicted target site (TS) positions for *miR-675-5p*. **b** (Left) Molecular beacon binding assay used to determine *miR-675-5p* binding to *DUX4*. Unbound, the molecular beacon folds into a stem loop structure that brings the ZenBHQ quencher in close proximity to a the 6FAM fluorophore, thereby preventing 6FAM fluorescence. We incorporated the mature sequence of *miR-675-5p* in the MB loop sequence. Hybridization of the MB to a complementary DUX4 target sequence separates the fluorophore and quencher, allowing fluorescence emission, which is quantified as a measure of binding. (Right) The graph shows binding of the mature *miR-675-5p* molecular beacon to target sites shown in (**a**). *miR-675-5p* bound 8 *DUX4* target sites (6 in the ORF and 2 in the 3'UTR). Six predicted TS did not bind to *miR-675-5p* (Supplementary Fig. 3b and Supplementary Data 1 for TS position and sequence). The TS neg. ctrl is a random sequence. Curves are fit into one site-specific binding with Hill slope equation. Each data point represents mean ± SD ($N = 3$ independent experiments). **c** Base-pairing of *miR-675-5p* and predicted *DUX4* target sites used in the molecular beacon assay. The binding affinity ($K_d$) of *miR-675-5p* molecular beacon to each target site was determined by subtracting background fluorescent signal from the molecular beacon signal (MBS), expressed in relative fluorescent units (RFU). The $K_d$ corresponds to the TS concentration (μM) required to reach half of maximum fluorescence. We generated RNA "mimic" bases in the *miR-675-5p*:TS pair and replaced "G" nucleotides with "A" nucleotides (in gray) whenever the "G" is facing a "T"[29]. Source data are provided as a Source data file.

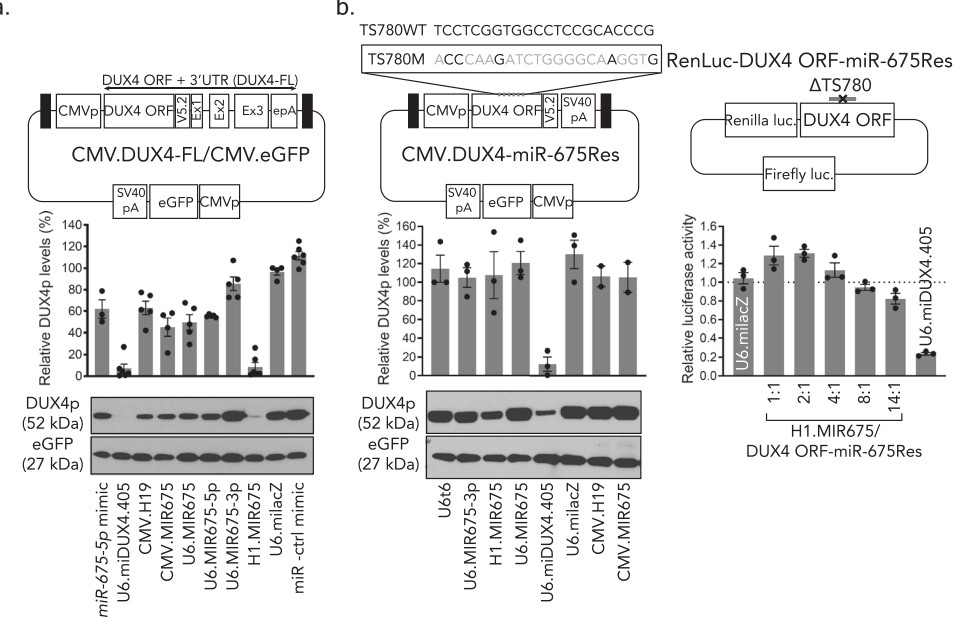

**Fig. 3 *miR-675-5p* specifically targets *DUX4* and reduces its protein and mRNA levels in vitro. a, b** Blinded western blots detecting V5 epitope-tagged DUX4 and an eGFP transfection control in the presence of *miR-675* or control constructs. **a** Proteins were harvested from HEK293 cells 48 h after co-transfection with the CMV.DUX4-FL/CMV.eGFP expression plasmid and indicated *miR-675* or control plasmids. Black boxes, AAV2 inverted terminal repeats (ITR). CMVp, cytomegalovirus promoter. Western blot membranes were stained with antibodies to the V5 epitope tag (to detect DUX4.V5) or eGFP. DUX4 protein was reduced in cells treated with CMV.H19 (35 ± 7%, $P = 0.0007$, $N = 5$ independent replicates), *miR-675-5p* mimic (36 ± 9%, $P < 0.0001$, $N = 3$), CMV.MIR675 (53 ± 9%; $P < 0.0001$, $N = 4$), U6.MIR675 (49 ± 8%; $P < 0.0001$, $N = 5$), and U6.MIR675-5p (42 ± 2%; $P = 0.0004$, $N = 5$). H1.MIR675 showed the highest DUX4 inhibition with an average of 91 ± 4% ($P < 0.0001$, $N = 6$), similar to U6.miDUX4.405 (93 ± 4%, $P < 0.0001$, $N = 6$). The U6.MIR675-3p construct did not significantly inhibit DUX4 protein levels (11 ± 7%, $P = 0.6286$, $N = 5$). **b** *Mir-675* specifically reduced *DUX4* expression via target engagement. *MiR-675*-resistant *DUX4* constructs expressing full-length DUX4 protein or the *Renilla* luciferase-DUX4 reporter (CMV.DUX4-miR-675Res and RenLuc-DUX4.ORF-miR-675Res) contain silent mutations in *miR-675* target site 780 (TS780) and lack *miR-675* binding sites in the *DUX4* 3′UTR. Western blots show that *miR-675* expression constructs did not significantly reduce *miR-675*-resistant DUX4 ($P = 0.9997$), while the U6.miDUX4.405 positive control, which had intact binding sites on *DUX4*, reduced mean DUX4 protein levels by 90 ± 6% ($P = 0.0003$). Similarly, increasing doses of the H1.MIR675 plasmid did not significantly silence the *miR-675*-resistant *Renilla* luciferase target ($P = 0.3467$ for the 14:1 *miR-675:DUX4 ORF-miR-675Res* ratio). Blots shown represent 6 (**a**) and 3 (**b**) independent blots (additional blots in Supplemental Data). DUX4 protein (DUX4p) was reported as mean % expression ± SEM and plotted above each representative image. Luciferase assay results (**b**) represent mean relative *Renilla* luciferase activity ± SEM, normalized to the miLacZ negative control ($N = 3$ independent experiments). For **a**, **b** one-way ANOVA followed by Dunnett's multiple comparison tests were performed for statistical analyses. TS780WT: wild-type TS780 sequence. TS780M: TS780 sequence with silent point mutations colored in gray. ΔTS780: deleted TS780. Source data are provided as a Source data file.

H1.MIR675 and CMV.H19 reduced Caspase 3/7 activity in cells co-transfected with CMV.DUX4 by 49 ± 1%, 34 ± 2%, and 24 ± 2% respectively ($P < 0.0001$, ANOVA, $N = 3$ for all three conditions) (Fig. 4b).

In addition to Caspase activation, DNA damage is another feature of apoptosis. Indeed, *DUX4* causes DNA damage that can be visualized and quantified using an alkaline comet assay (single cell electrophoresis)[36,37]. To corroborate the findings from the Caspase-3/7 assay, we performed comet assays to assess DUX4-mediated DNA damage in the presence and absence of *miR-675*. To do this, we co-transfected HEK293 cells with plasmids expressing *DUX4* and a negative control U6.milacZ; *DUX4* and H1.MIR675; or *DUX4* and U6.miDUX4.405 positive control (Fig. 5a). Additional controls included mock-transfected cells or cells transfected with a plasmid expressing our previously described, non-toxic *DUX4.HOX1* construct that contains mutations in the first DNA binding domain[13]. In cells co-expressing *DUX4* and miLacZ, DNA damage was evident by abundant DNA comet tails representing DNA fragments migrating away from cell nuclei (Fig. 5a). In these cells, the tail moment reached an average of 30 ± 9 μm. In contrast, cells lacking *DUX4* or expressing the non-toxic *DUX4.HOX1* mutant showed no DNA fragmentation with an average tail moment of 0.67 ± 0.15 μm and

0.28 ± 0.10 μm, respectively. Importantly, co-expression of *DUX4* with either *miDUX4.405* positive control or H1.MIR675 prevented DUX4-associated DNA fragmentation as the average tail moment reached 5 ± 4 μm for both constructs (~77% reduction when compared to DUX4 + milacZ transfected cells). These results confirmed that DUX4 inhibition by the *miDUX4.405* positive control and *miR-675* protected cells from apoptosis.

**miR-675 inhibits transactivation of a DUX4-responsive reporter.** DUX4 binds DNA and transactivates numerous target genes involved in pathways including oxidative stress and cell death, among others. To investigate more directly the ability of *miR-675* and its precursor *H19* to inhibit DUX4 transactivation activity, we utilized a previously described reporter construct containing five DUX4-specific binding sites upstream of GFP[38]. We co-transfected HEK293s with the DUX4-responsive GFP reporter and plasmids expressing DUX4- and either *milacZ*, *H19*, or *miR-675*, then measured mean GFP fluorescence intensity (MFI) using flow cytometry (Fig. 5b). GFP MFI was reduced 49 ± 2% and 57 ± 2% in cells expressing *H19* or *miR-675*, compared to cells treated with the U6.miLacZ negative control (Fig. 5b).

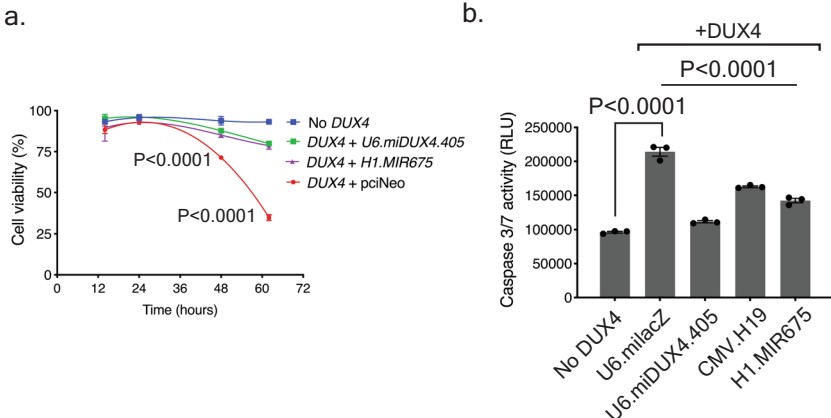

**Fig. 4 miR-675 protects cells from DUX4-induced cell death. a** Cell viability assay in HEK293 cell transfected with indicated plasmids. Cell death was measured by counting Trypan blue-stained cells over the indicated time course and calculated by subtracting the number of dead cells from the total number of cells plated per well. No significant difference in cell viability was seen before 36 h of transfection. Cells co-transfected with *DUX4* and the control pciNeo plasmid showed 23 ± 2% (*P* < 0.0001) and 58 ± 1% (*P* < 0.0001) cell death by 48 and 62 h, respectively, when compared to the mock-transfected cells (No *DUX4*). However, co-transfection with *DUX4* and H1.MIR675 or U6.miDUX4.405 protected HEK293 cells from DUX4-induced cell death, as only 9 ± 2% and 6 ± 2% of cells were dead by 48 h, respectively, and slightly increased by 62 h. These values were not significantly different from mock-transfected HEK293s but represented significant protection from cell death compared to the DUX4 + pciNeo control (*P* < 0.0001 for both H1.MIR675 and U6.mi405). Two-way ANOVA followed by Tukey's multiple comparison tests were performed for statistical analyses. Results were reported as the mean cell viability ratio ± SEM (*N* = 3 independent experiment). **b** Caspase-3/7 assay 48 hrs after co-transfection of HEK293 cells with *DUX4* expression plasmid and either U6.miLacZ, U6.miDUX4.405, H1.MIR675, or CMV.H19. The U6.miLacZ negative control failed to protect cells from Caspase-3/7 activation (2.2 ± 0.1-fold increase; *P* < 0.0001) compared to mock-transfected cells. In contrast, Caspase-3/7 activity was significantly decreased in cells transfected with *DUX4* and U6.miDUX4.405, H1.MIR675 and CMV.H19 (49 ± 1%, 34 ± 2% and 24 ± 2%, respectively; *P* < 0.0001). Two-way ANOVA followed by Dunnett's multiple comparison tests were performed for statistical analyses. Results reported as mean relative luminescence units (RLU) ± SEM (*N* = 3 independent experiment). Source data are provided as a Source data file.

**Endogenous *miR-675* targets *DUX4* sequences in human FSHD myotubes**. To this point in our study, we used transfected HEK293 cells to test the hypothesis that *miR-675* inhibits *DUX4* expression. Although HEK293s hold several advantages for this type of work, they do not normally express polyadenylated *DUX4* or *miR-675* and are not muscle cells. As a next step, we sought to determine if *miR-675* could regulate *DUX4* expression in human muscle cells that naturally expressed *DUX4* and/or *miR-675*. Accomplishing this required two new sets of reagents: (i) antagomirs, which are small oligonucleotides used to silence endogenous miRNAs; and (ii) three human myoblast cell lines, including two from FSHD-affected patients (15A and 17A) that express sporadic and low levels of endogenous *DUX4*; and an unaffected, *DUX4*-negative control myoblast line (15V)[39,40]. Prior to using these reagents experimentally, we opted to first validate their functionality by confirming *miR-675* and *DUX4* gene expression in 15A, 17A, and 15V cells (Supplementary Note 5, Supplementary Fig. 10 and Supplementary Table 1). We also empirically tested the impact of inhibiting *miR-675* with an antagomir on *Renilla* luciferase activity encompassing in its 3′ UTR a perfect *miR-675-5p* target site (RenLuc-miR-675R), and on *CDC6*, a known *miR-675* target gene (Supplementary Note 5, Supplementary Figs. 11 and 12).

We next examined the ability of *miR-675* inhibition to de-repress wild-type and mutant *DUX4* sequences. We again employed the dual luciferase reporters, where one plasmid contained full-length wild-type *DUX4* sequences (DUX4-FL) and a second contained *miR-675*-resistant mutations (DUX4-ORF-mir675Res). These plasmids were co-transfected into 15V unaffected and 15A FSHD myoblasts along with *miR-675* or negative control antagomirs (anti-miR-675 or anti-miR-ctrl). The presence of anti-miR-675 significantly increased the relative *Renilla* luciferase activity from RenLuc-DUX4-FL in 15V cell lines by 1.5 ± 0.1-fold (*P* < 0.0001, ANOVA, *N* = 3) and in 15A

cell lines by 2.1 ± 0.1-fold (*P* < 0.0001, ANOVA, *N* = 3), while the negative control antagomir had no impact (Fig. 6a). These results suggested that *miR-675* was able to silence *DUX4* sequences via RNAi. To confirm these results, we performed the same experiment using *Renilla* luciferase tagged with *miR-675*-resistant *DUX4* sequences. In all cases, cells expressing the *miR-675*-resistant plasmid produced more luciferase activity, compared to their counterparts expressing *Renilla* luciferase fused to the wild-type *DUX4* sequence (Fig. 6a). Nevertheless, we still saw de-repression when anti-miR-675 was transfected along with the *miR-675* resistant luciferase plasmid (1.8- to 1.3-fold increase in luciferase activity, *P* < 0.0001 and *P* = 0.0078, respectively in 15V and 15A cells compared to cells that received the anti-miR-ctrl). These results suggested that *miR-675* was still capable of mediating some inhibition, albeit weaker, on the resistant *Renilla* luciferase-*DUX4* fusion sequence by binding to the remaining, non-mutated, lower affinity target sites on the *DUX4* sequence.

Using QPCR, we found that the anti-miR-675 antagomir significantly reduced *miR-675-5p* by an average of 23 ± 2-fold (*P* < 0.0001, ANOVA, *N* = 3 independent experiments) (Fig. 6b). In addition, 15A FSHD cells transfected with anti-miR-675, and subsequently differentiated for 4 days, had levels of the DUX4-responsive biomarkers *TRIM43* and *KHDC1L* increase by 3-fold when compared to cells that received the anti-mir control (*P* < 0.0001, ANOVA, *N* = 3 independent experiments). In a second anti-miR study in differentiated 15 A cells, we measured *DUX4*, *miR-675-5p*, and *TRIM43* expression using ddPCR. *MiR-675-5p* expression significantly decreased (12 ± 1-fold; *P* = 0.0118, ANOVA, *N* = 3 independent experiments), while endogenous *DUX4* and the DUX4-responsive biomarker *TRIM43* significantly increased by 3.4 ± 0.6-fold (*P* < 0.0001, ANOVA, *N* = 3 independent experiments) and 1.9 ± 0.3-fold (*P* = 0.0074, ANOVA, *N* = 3 independent experiments), respectively (Fig. 6b). These data are consistent with expected results arising from inhibiting a *DUX4* inhibitor (anti-

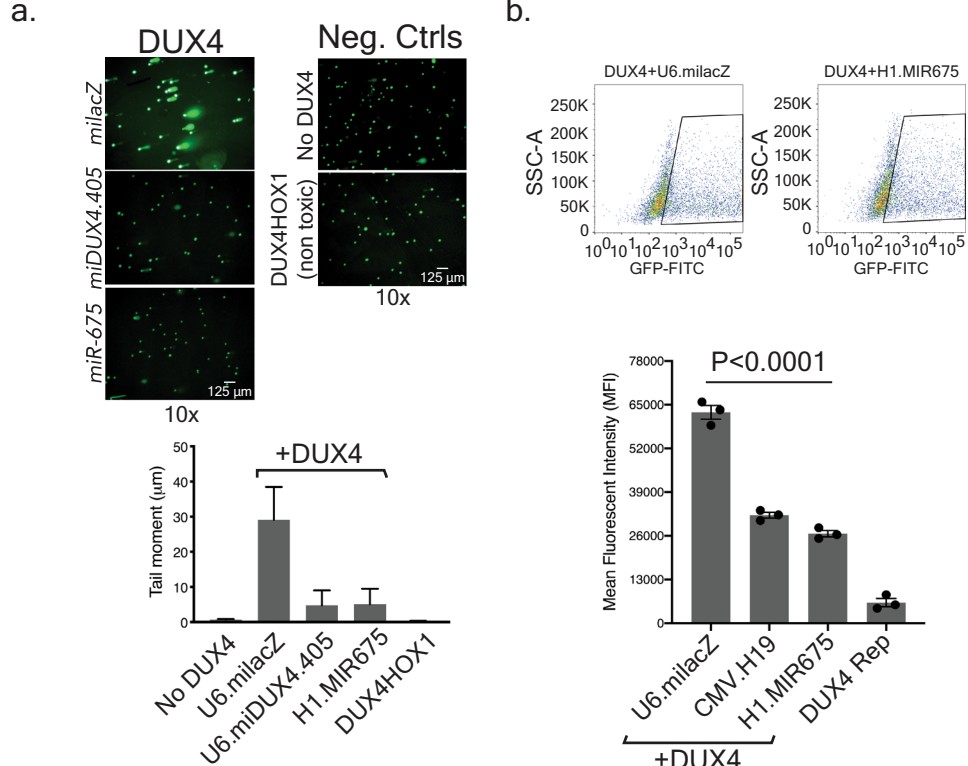

**Fig. 5 *miR-675* protects cells from DUX4-induced DNA damage and transactivation. a** Alkaline comet assay using HEK293 cells co-transfected with *DUX4* and U6.milacZ, H1.MIR675, U6.miDUX4.405, or negative controls, including the non-toxic DUX4.HOX1 mutant. Comet-like tails in electrophoresed cells indicate DNA damage. To quantify DNA damage, cells were stained with SYBR green fluorescent dye to detect DNA fragments and Tritek CometScore software was used to analyze 10 images of randomly selected non-overlapping cells from two independent replicates. In each image, the average tail moment was calculated (tail length × percent tail DNA) from 30 to 60 nuclei. Mean tail moment was 30 ± 9 μm in *DUX4*-expressing cells treated with the miLacZ negative control, while negative controls showed no DNA damage (0.67 ± 0.15 μm and 0.28 ± 0.10 μm, for No DUX4 or DUX4.HOX1, respectively). In contrast, U6.miDUX4.405 and H1.MIR675 protected cells from DUX4-induced DNA damage (mean tail moment of 4.75 ± 4.25 μm and 5.11 ± 4.38 μm, respectively). Data in graph represent the mean tail moment (μm) ± SEM of two independent transfection replicates, repeated with similar results. **b** *Mir-675* reduced DUX4 transactivation function in a DUX4-activated GFP reporter assay. HEK293 cells were co-transfected with *DUX4* and either U6.milacZ, CMV.H19 or H1.MIR675, as well as a DUX4-activated GFP reporter (DUX4 Rep), which contains five DUX4-specific binding sites[38]. Mean fluorescence intensity (MFI) was measured using flow cytometry. Cells co-transfected with *DUX4*, U6.milacZ and the DUX4-activated GFP reporter showed a MFI of 62744 ± 2040, while CMV.H19 or H1.MIR675 reduced GFP expression by 49 ± 2% (32148 ± 865 MFI; *P* < 0.0001) and 57 ± 2% (26667 ± 930 MFI; *P* < 0.0001), respectively. One-way ANOVA followed by Dunnett's multiple comparison tests were performed for statistical analyses. Values are given as mean MFI ± SEM (*N* = 3 independent experiment). Source data are provided as a Source data file.

miR-675 =↓ *miR-675* =↑ DUX4 = ↑ *DUX4*-responsive biomarkers) (Fig. 6). Importantly, using the most comprehensive atlas of miRNA-target interactions (miRwalk2.0), *miR-675* was not predicted to target *TRIM43* nor *KHDC1L*, suggesting that their expression changes indirectly resulted from *miR-675* knockdown of *DUX4*[41] (Supplementary Data 2, 3, and 4).

**Endogenous *miR-675* regulates DUX4 protein expression in myotubes**. To evaluate the effect of endogenous *miR-675* on the levels of transfected *DUX4*, we performed western blot on proteins extracted from 15 A FSHD cells transfected with V5-tagged *DUX4-FL* or *miR-675*-resistant *DUX4* (DUX4-miR-675Res) expression plasmids with and without anti-miR-675. After four days of differentiation, we detected minimal amounts of DUX4.V5 protein in one of the three replicates transfected with DUX4 alone (Fig. 7a and Supplementary Fig. 13). No DUX4-signal was found in the other two replicates. We found increased DUX4 signal upon adding anti-miR-675 (Fig. 7a and Supplementary Fig. 13). DUX4 signal was notably higher in cells transfected with the *miR-675*-resistant *DUX4* construct, and some additional de-repression was evident in cells treated with anti-miR-675, suggesting that *miR-675* could still repress *DUX4*,

perhaps utilizing the minor binding sites we did not mutate in the resistant constructs. The difference in DUX4 protein levels in cells transfected with the wild-type and *miR-675*-resistant construct was surprisingly large. We engineered the *miR-675*-resistant *DUX4* construct to contain silent mutations in the TS780 *miR-675* binding site and to not contain its 3'UTR sequence and, therefore, expresses wild-type DUX4 protein. To rule out the possibility that unexpected differential toxicity could contribute to the differences in DUX4 protein levels arising from the two constructs, we used Caspase-3/7 assay to confirm that both sequences were capable of producing cell death (Supplementary Note 6 and Fig. 7a). As a result, DUX4 levels in the western blots correlated with Caspase-3/7 activation, as cells with the highest amounts of DUX4 had increased levels of cell death. In this Caspase-3/7 assay, we also showed that the inhibition of *miR-675* with anti-miR-675 increased cell death in 15A FSHD myotubes.

Next, to evaluate the effect of endogenous *miR-675* on endogenous DUX4, we transfected 15A FSHD myoblasts with anti-miR-675 or anti-miR -ctrl, differentiated cells for 4 days, and then used immunofluorescence to detect DUX4 protein. Anti-miR-675-treated myotubes showed a 7 ± 2-fold increase in DUX4+ nuclei compared with myotubes treated with anti-miR-

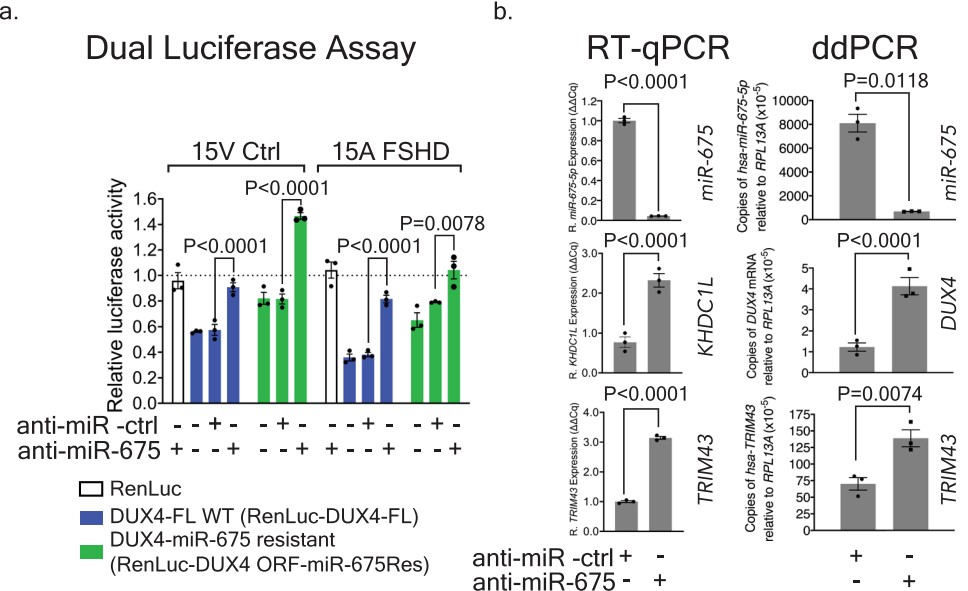

**Fig. 6 Endogenous *miR-675* targets *DUX4* and prevents expression of DUX4-responsive biomarkers in 15A FSHD-affected human myotubes. a** Dual-luciferase assay in 15V Ctrl and 15A FSHD myoblasts. *DUX4*-tagged *Renilla* luciferase (RenLuc-DUX4-FL) or *miR-675*-resistant *Renilla* luciferase (RenLuc-DUX4 ORF-miR-675Res) plasmids were co-transfected with 300 nM anti-miR- ctrl or anti-miR-675 into 15V and 15A cells. Relative *Renilla* luciferase activity was measured at 48 h. Addition of *DUX4* sequences to the end of *Renilla* luciferase caused a significant decrease in relative luciferase activity in both cell lines (compare RenLuc-DUX4-FL to RenLuc only; mean 41 and 65% reduction in luciferase activity in 15V and 15A cells respectively; ($P < 0.0001$). Transfection of the anti-miR-675 caused a 1.5–2.0-fold increase of luciferase activity in the DUX4-FL-tagged *Renilla* luciferase construct, compared to cells that received the anti-miR-ctrl ($P < 0.0001$). All data represent mean normalized *Renilla* luciferase activity ± SEM ($N = 3$ independent experiments). Two-way ANOVA followed by Tukey's multiple comparison tests were performed for statistical analyses. **b** Gene expression results in 4-day differentiated 15A FSHD cells transfected with 300 nM anti-miR-675 or an anti-miR -ctrl. Left panels, QPCR. Treatment with anti-miR-675 decreased mean *miR-675-5p* levels 23 ± 2-fold ($P < 0.0001$) and caused a 3-fold increase of *DUX4*-responsive biomarkers TRIM43 and KHDC1L3 ($P < 0.0001$). Results were reported as gene expression ($\Delta\Delta Cq$) ± SEM ($N = 3$ independent experiments), normalized to 15A FSHD cells transfected with *anti-mir-ctrl*. Right panels, ddPCR results for endogenous *DUX4*, *miR-675*, and *TRIM43* using RNA acquired from a similar set of anti-mir transfection experiments. Treatment with anti-miR-675 decreased mean *miR-675-5p* levels 12 ± 1-fold ($P = 0.0118$), while *DUX4* and TRIM43 expression increased 3.4 ± 0.6-fold ($P < 0.0001$) and 1.9 ± 0.3-fold ($P = 0.0074$), respectively. Results were reported as the mean absolute *DUX4*, *miR-675-5p*, or *TRIM43* concentration (copies/μL) ± 95% poisson confidence interval normalized to human *RPL13A*. Data represent mean relative concentration (copies/μL) ± SEM ($N = 3$ independent experiments) normalized to cells transfected with *anti-mir-ctrl*. For QPCR and ddPCR, two-way ANOVA followed by Sidak's multiple comparison tests were performed for statistical analyses. Source data are provided as a Source data file.

ctrl ($P = 0.0369$, unpaired two-tailed *t*-test, $N = 3$ independent experiments) (Fig. 7b). Taken together, these results suggested that the endogenous *miR-675* targeted the transfected or endogenous *DUX4* sequence and reduced DUX4 protein levels, while *DUX4* expression and DUX4-induced toxicity were increased upon *miR-675* inhibition with the anti-miR-675 antagomir.

**AAV-based *miR-675* gene therapy reduces *DUX4* expression and counteracts DUX4-induced toxicity in a *DUX4* mouse model**. Our results to this point suggested that *miR-675* upregulation might be useful as an FSHD therapy. To test this idea, we used two strategies: (i) AAV.MIR675-based gene therapy in a *DUX4*-expressing mouse model; and (ii) drug-based therapy aimed to increase *miR-675* expression in FSHD patient cells. For gene therapy, we packaged *miR-675* expression cassettes into serotype 6 self-complementary adeno-associated viral vectors (scAAV6), which are myotropic. Unfortunately, the original H1.MIR675 sequence we used for in vitro studies caused histopathology when delivered to wild-type C57BL/6 mouse muscle in vivo with AAV (Supplementary Fig. 14). The inability to safely translate a microRNA expression cassette from an in vitro to in vivo system was not unprecedented[29], and could result from misfolding of 5′ and 3′ sequences flanking the pre-miRNA, leading to suboptimal processing in vivo. We, therefore,

optimized these sequences to improve folding, while maintaining the natural *pre-mir-675* sequences, and identified a construct capable of producing mature *miR-675* without causing in vivo toxicity in mouse muscle (*optimization manuscript in preparation*). We next tested the ability of this new vector (scAAV6.-MIR675) to protect mouse muscles from DUX4-induced damage in vivo using an AAV-based *DUX4* model similar to one we previously described, but containing the *DUX4* 3′UTR sequences in addition to a V5 epitope-tagged *DUX4* open reading frame (ORF)[13,42]. We used this construct because *miR-675* possesses target sites within the *DUX4* ORF and 3′UTR (Fig. 2). We co-injected $5 \times 10^{10}$ scAAV6.MIR675 particles, or $5 \times 10^{10}$ particles of the non-targeting AAV.miLacZ control, with a $3 \times 10^9$ AAV6.CMV.DUX4-FL, into C57BL/6 mouse TA muscles (Fig. 8 and Supplementary Fig. 15). We then assessed DUX4-induced muscle histopathology, and measured the expression of *DUX4* and DUX4-responsive mouse biomarkers.

Muscles co-injected with AAV.milacZ and AAV.CMV.DUX4-FL showed obvious and widespread histopathology, with mononuclear cell infiltration, and an abundance of centrally nucleated myofibers (mean 24 ± 5%), a feature which indicates muscle damage and repair. In contrast, muscles co-injected with scAAV6.MIR675 with and without AAV.CMV.DUX4-FL were histologically normal (mean 5 ± 1% centrally nucleated myofibers), demonstrating that scAAV6.MIR675 is not toxic to normal

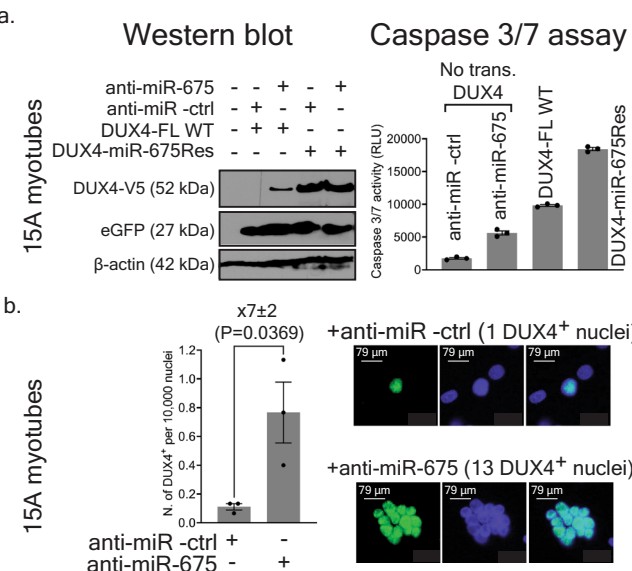

**Fig. 7 Endogenous *miR-675* targets *DUX4* and prevents DUX4-induced toxicity in 15A FSHD-affected human myotubes. a** Western blots. V5-epitope tagged wild-type or miR-675-resistant DUX4 plasmids were transfected into 15A FSHD myotubes co-treated with anti-miR-675 or anti-miR-ctrl. DUX4 expression plasmids also contained a separate CMV.eGFP reporter that was used as a transfection control and western blot normalizer. Proteins were harvested 4 days after differentiation, followed by western blot using antibodies to detect V5, eGFP, and β-actin. DUX4-V5 signal was absent in mock-transfected cells and below the level of detection in cells transfected with wild-type *DUX4* and the anti-miR-ctrl. However, DUX4.V5 signal was detected in cells co-transfected with wild-type *DUX4* and anti-miR-675. Increased DUX4 protein was evident in cells transfected with *miR-675*-resistant DUX4, using both anti-miR-675 and anti-miR-ctrl. Two additional western blot replicates were performed (Supplementary Fig. 13). In Caspase-3/7 assay, anti-miR-675 transfection increased Caspase-3/7 activity in 15A cells by 3.2 ± 0.3-fold compared to those treated with anti-miR-ctrl ($P < 0.0001$), while with DUX4 full-length or miR-675-resistant DUX4 expression plasmids increased Caspase-3/7 activity by 5.6 ± 0.4-fold and 10.6 ± 0.8-fold, respectively ($P < 0.0001$). Results were reported as the mean Caspase 3/7 activity (RLU) ± SEM ($N = 3$ independent experiments). One-way ANOVA followed by Tukey's multiple comparison tests were performed for statistical analyses. **b** Immunofluorescence (IF) to detect endogenous DUX4 protein in 4-days differentiated 15A FSHD myotubes transfected with 150 nM of anti-mir-ctrl or anti-miR-675. Cells were stained with a 1:100 dilution of DUX4 antibody (mouse mAB 9A12) followed by fluorescence-labeled secondary antibodies. Myotubes treated with anti-mir- ctrl and anti-miR-675 had an average of 0.11 ± 0.02, and 0.77 ± 0.21 DUX4+ per 10,000 nuclei, respectively. Myotubes treated with anti-miR-675 had their number of DUX4+ nuclei increase by 7 ± 2-fold ($P = 0.0369$). The graph shows quantification of number of DUX4+ per 10,000 nuclei. Results were reported as DUX4+ nuclei per 10,000 nuclei ± SEM ($N = 3$ independent experiments, repeated with similar results with two technical replicates per experiment). Unpaired two-tailed *t* test was performed for statistical analyses. Source data are provided as a Source data file.

mouse muscle and, importantly, can counteract DUX4-induced muscle damage in vivo (Fig. 8 and Supplementary Fig. 15). Consistent with the histopathological protection, compared to controls, scAAV6.MIR675-injected muscles also showed significantly reduced expression of *DUX4* (56 ± 12%, $P = 0.01$) and the *DUX4*-responsive biomarkers *Trim36* and *Wfdc3*, (88 ± 4%, $P = 0.0004$; and 57 ± 13%, $P = 0.011$, respectively), as determined by ddPCR (Fig. 8a). We also measured V5-tagged DUX4 protein

levels using western blot, and found less DUX4 protein in AAV.DUX4-injected TA muscles co-injected with scAAV6.-MIR675 compared to AAV.miLacZ controls (Fig. 8a and Supplementary Fig. 16). Finally, we stained TA muscle sections from co-injected animals with a red fluorophore-tagged anti-V5 antibody to detect V5-tagged DUX4 protein, and found reduced or absent DUX4 protein expression in TA muscle sections treated with AAV.CMV.DUX4-FL and scAAV6.MIR675, compared to those that received identical amounts of AAV.CMV.DUX4-FL and non-targeting miLacZ control vectors (Fig. 8b and Supplementary Fig. 17). As previously reported using the AAV.DUX4 model, some damaged muscle fibers showed cytoplasmic staining of DUX4 protein. We speculate this is due to nuclear membrane leakage in dying myofibers (Fig. 8b).

**Small molecule upregulation of *miR-675* reduces *DUX4* and *TRIM43* expression in FSHD patient myotubes.** Gene therapy has the potential to provide stable and long-lasting expression of a therapeutic microRNA product, such as *miR-675*. However, there are also potential risks; if a vector-delivered microRNA causes unforeseen off-target effects, it is currently not possible to turn it off or tune down expression. We therefore also pursued an alternative strategy that, to our knowledge, has never been previously attempted. Specifically, we sought to increase endogenous *miR-675* expression in FSHD affected muscle cell lines using a drug-based therapy, which would be tunable and potentially stopped if untoward events arose. Having identified *miR-675* as a strong endogenous regulator of *DUX4*, we decided to search within previously published gene expression data for small molecules that increase *miR-675* expression or its *H19* precursor. We identified three candidates (β-estradiol, combination of β-estradiol + medroxyprogesterone acetate (MPA), and melatonin) that were shown to increase *miR-675* expression in various models but had not been tested in the cell models we used here, HEK293s and human myotubes[43–45]. We first confirmed the system in transfected HEK293s (Supplementary Note 7, Supplementary Data 5, and Supplementary Fig. 18). We then tested the effect of the three molecules on the expression of endogenous *miR-675-5p*, *DUX4* and DUX4-responsive biomarker, *TRIM43* in 15A, 18A, and 17A FSHD differentiated muscle cell lines (myotubes), which express different levels of DUX4 (15A < 18A < 17A) (Fig. 9)[39]. All three molecules were added to myotubes on the 4th day of differentiation and cells were harvested 24 h later. In 15A myotubes, all three molecules triggered an increase in *miR-675* expression (Supplementary Table 2 and Fig. 9a). *DUX4* and *TRIM43* mRNA expression significantly decreased in 15A myotubes treated with β-estradiol+MPA or melatonin. However, the treatment with β-estradiol alone did not trigger a significant decrease in *TRIM43* expression (Supplementary Table 2 and Fig. 9a). In 17A and 18A myotubes, β-estradiol, β-estradiol +MPA, or melatonin triggered a significant increase in *miR-675* expression that was associated with significant decreases in *DUX4* and *TRIM43* mRNA expression (Supplementary Table 2 and Fig. 9a). Importantly, control experiments suggested that these drugs did not directly reduce *TRIM43* or significantly impact myoblast differentiation, thereby supporting our hypothesis that RNAi mediated by small molecule upregulation of *miR-675* could reduce *DUX4* and indirectly decrease a DUX4-responsive biomarker in FSHD muscle cells (Supplementary Note 8 and 9 and Supplementary Fig. 19).

Finally, to assess the effect of the three molecules on DUX4 protein levels, we performed an immunofluorescence assay in 18A FSHD affected myotubes differentiated for 4 days[39]. Myotubes were treated with β-estradiol, β-estradiol, and MPA, melatonin, or vehicle, and then stained for DUX4 protein 24 h

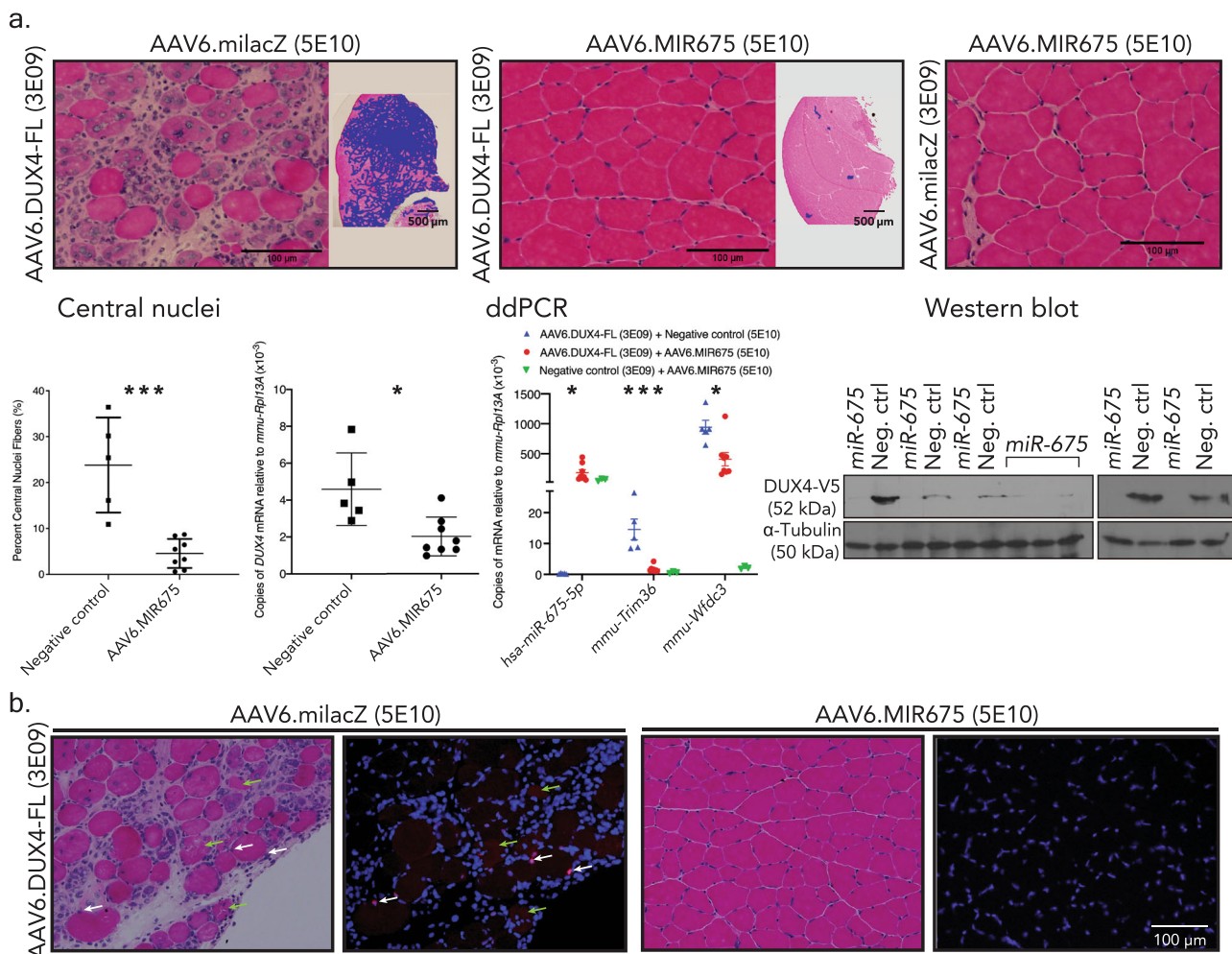

**Fig. 8 *miR-675* protects mouse skeletal muscles from DUX4-induced muscle damage. a** H&E staining, central nuclei counts, and gene expression analysis of AAV-injected adult mouse (C57BL/6) tibialis anterior (TA) muscles 2 weeks after IM delivery of indicated vectors. Images show 10 μm H&E-stained cryosections at high (×20) and low power (×4). To visualize the breadth of lesions on low-power images, fibers with central nuclei (CN), degeneration or inflammation were intentionally shaded with a purple digital overlay. Control-treated, DUX4-expressing muscles show histopathological evidence of degeneration, including myofibers with inflammatory infiltrates, central nuclei, and variable fiber size (top left). In contrast, the AAV6.MIR675 vector protected muscles from DUX4-induced damage at the same time point. High-power photos show representative images at indicated vector dosages. Images are representative of the following N's: AAV6.DUX4-FL + AAV6.MIR675, N = 8 TA muscles; AAV6.DUX4-FL + negative control (AAV6.miLacZ or AAV6.eGFP), N = 5 TA muscles; No *DUX4*, i.e., AAV6.miLacZ + AAV6.MIR675, N = 3 TA muscles. CN counts from 20x H&E-stained TA muscle serial sections. (Left) AAV6.MIR675-treated muscles showed 81 ± 6% fewer myofibers with central nuclei compared to the negative control-treated muscles (****P = 0.0004, two-tailed unpaired *t*-test). Droplet digital PCR (ddPCR) showed significant 56 ± 12% reduction in *DUX4* mRNA in AAV6.MIR675-treated versus negative control-treated TA muscles (N = 8 *miR-675*-treated and N = 5 negative control-treated mice; unpaired two-tailed *t* test, *P = 0.01). (Middle) ddPCR of *miR-675* and *DUX4*-responsive mouse biomarkers in AAV6.MIR675 treated and un-treated TA muscles. *Trim36* and *Wfdc3* expression was reduced by 88 ± 4% (***P = 0.0004) and 57 ± 13% (*P = 0.011), respectively. Results reported as average absolute *DUX4, hsa-miR-675-5p, Trim36* or *Wfdc3* copies/μL ± 95% poisson confidence interval ± SEM, normalized to *mmu-Rpl13A*. For *miR-675-5p, Wfdc3*, and *Trim36*, one-way ANOVA followed by Tukey's multiple comparison tests were performed for statistical analyses. (Right) Western blots using proteins from injected TA muscles (uncropped western blots in Supplementary Fig. 16). Anti-V5 epitope antibodies were used to detect V5-tagged DUX4, with α-tubulin used as a loading control. **b** High-power photos show representative images at indicated vector dosages of N = 8 TA co-injected with AAV6.MIR675 or negative control vector with AAV6.DUX4-FL. H&E or immunofluorescence staining of 10 μm cryosections, using antibodies to detect DUX4 (V5, red) or DAPI (nuclei, blue). White arrows indicate myonuclei showing DUX4-V5 signal (red). Green arrows indicate degenerating myofibers containing some red signal in the cytoplasm, potentially indicating nuclear breakdown and DUX4 leaking into the cytoplasm[21]. All Ns were repeated with similar results.

later using a DUX4-targeting antibody (mouse mAb 9A12, 1:100 dilution). Similar to previous reports, DUX4 protein was rare, as mock-treated myotubes had an average of 12.8 ± 0.8 DUX4+ per 10,000 nuclei[39]. However, DUX4+ nuclei decreased in myotubes treated with β-estradiol, β-estradiol and MPA or melatonin by 35 ± 5%, 48 ± 5%, and 54 ± 4%, respectively (8.3 ± 0.4, 6.7 ± 0.5, and 6.0 ± 0.4 DUX4+ nuclei per 10,000) (Fig. 9b).

## Discussion

The discovery of RNAi and mammalian microRNAs over 20 years ago added to the diversity of functions that RNA molecules play in gene expression, and importantly, provided new tools to develop powerful gene silencing therapies. Most RNAi therapy strategies use artificial RNAi triggers, like siRNAs, shRNAs, or microRNA shuttles, which are engineered to contain ~22

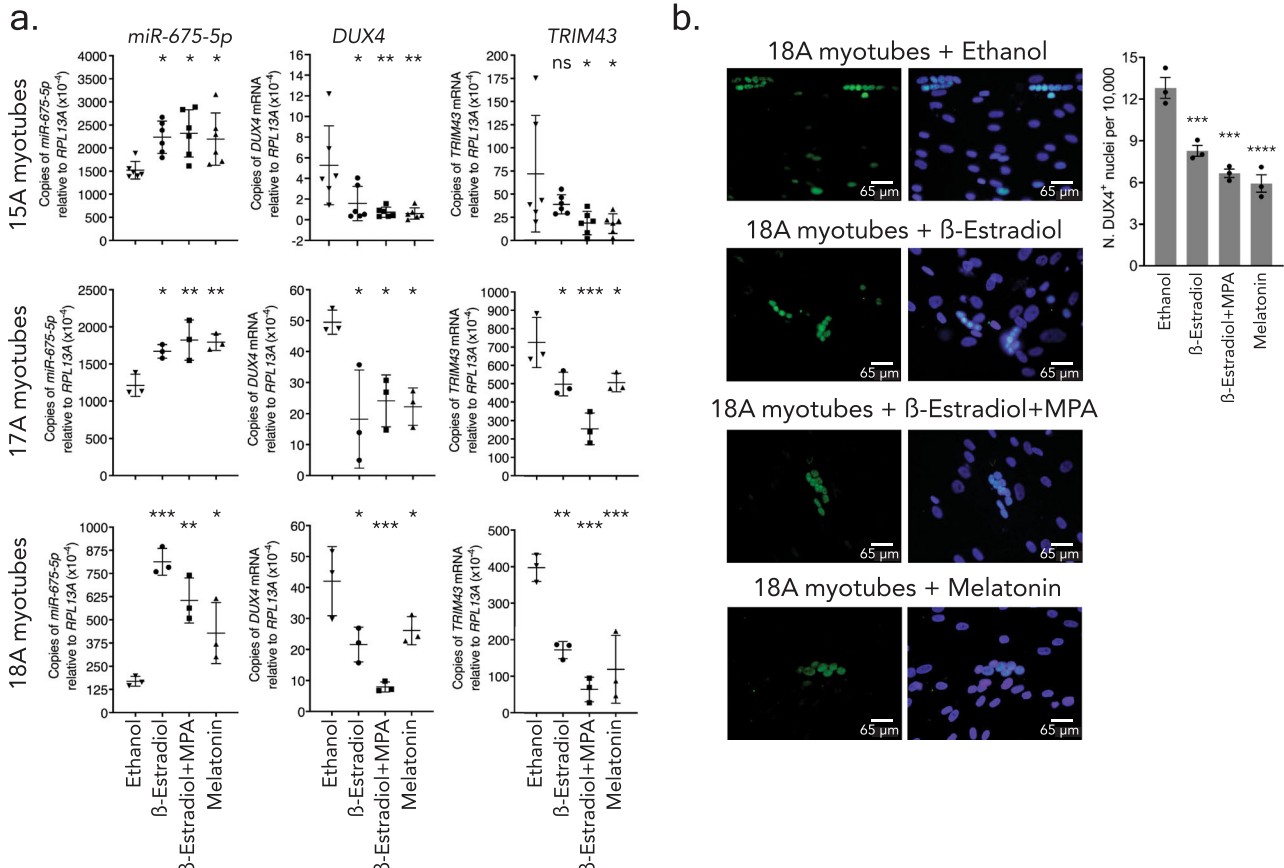

**Fig. 9 β-estradiol, medroxyprogesterone acetate (MPA), and melatonin increased *miR-675* expression and reduced the expression of *DUX4* and DUX4-responsive biomarker *TRIM43* in 3 FSHD affected myotube lines. a** Droplet digital PCR (ddPCR) to measure *miR-675-5p*, *DUX4*, and *TRIM43* levels in 15A, 17A, and 18A FSHD affected myotubes, 5 days after differentiation. We added two drugs individually (i.e., β-estradiol and melatonin) or as a combination (β-estradiol + MPA) on the 4th day of differentiation. All treatments were compared to ethanol (vehicle) treated control cells. $N = 6$ independent experiments for 15A and $N = 3$ for 17A and 18A. *$P = 0.05$. **$P = 0.01$. ***$P = 0.001$. Results were reported as the average absolute *DUX4*, *miR-675-5p*, or *TRIM43* concentration (copies/μL) ± 95% poisson confidence interval normalized to *hsa-RPL13A*. Data represent mean relative concentration (copies/μL) ± SEM ($N = 3$–6 independent experiments). **b** Immunofluorescence on 18 A FSHD affected myotubes following 5 days of differentiation, and 24 h post-treatment with 20 μM of β-estradiol, β-estradiol, and MPA or Melatonin. DUX4 protein was detected with 1:100 of DUX4 antibody (mouse mAb 9A12). Myotubes mock-treated with 100% ethanol had an average of 12.8 ± 0.8 DUX4+ per 10,000 nuclei. Treatment with β-estradiol, β-estradiol and MPA or Melatonin reduced DUX4+ nuclei per 10,000 in myotubes (8.3 ± 0.4, 6.7 ± 0.5 and 6.0 ± 0.4 DUX4+ nuclei per 10,000, respectively; ***$P = 0.0010$; ***$P = 0.0001$; ****$P < 0.0001$). Results represent mean DUX4+ nuclei per 10,000 nuclei ± SEM ($N = 3$ independent experiments, repeated with similar results with two technical replicates per experiment). For **a**, **b** one-way ANOVA followed by Dunnett's multiple comparison tests were performed for statistical analyses.

nucleotide (nt) of perfect sequence complementarity with a target gene[46]. Indeed, to date the only three FDA-approved RNAi drugs, Alnylam's Patisiran, Oxlumo and Givlaari, utilize an engineered siRNA approach to inhibit disease genes. Similarly, our published *DUX4*-targeting gene therapy strategy utilizes an engineered microRNA shuttle system in which the mature sequences have perfect 22 nt base-pairing with the *DUX4* transcript. The main advantage of this approach is potency; in general, greater complementarity between a target transcript and a mature microRNA yields greater gene silencing, often >90%. In contrast, most natural microRNAs have several mismatches with their gene targets, leading to relatively weak or modest silencing.

Despite the advantages of using an engineered system, some studies have described using natural microRNA sequences as potential treatments for disease, such as *mir-26a* gene replacement in a mouse model of *mir-26a*-deficient hepatocellular carcinoma[47]. In all cases described to date, these natural microRNA sequences were exogenously delivered as oligonucleotides or using gene therapy vectors. Here, we also pursued a

strategy to use a natural microRNA—*miR-675*—for therapeutic purposes. In addition to using a traditional gene therapy approach to deliver *miR-675* (Fig. 8), we also developed a small molecule-based strategy that, to our knowledge, had never been described before for any disease (Fig. 9). Specifically, we pursued a simple therapeutic model based on two main ideas: (i) endogenous microRNA gene expression can change in response to small molecule treatments; (ii) if natural *DUX4*-targeting microRNAs exist, their upregulation should serve to decrease *DUX4* expression via the RNAi pathway. By combining these two ideas, we reasoned that an FSHD therapy could be developed by using small molecules to increase expression of natural microRNAs that target *DUX4* for degradation within the cell. Thus, our approach was target-based and relied upon a known mechanism of *DUX4* inhibition with RNAi. From the outset, there were two practical problems with this strategy: (i) there were no microRNAs known to regulate *DUX4*; and (ii) even if we could identify a *DUX4*-targeting microRNA, we were not certain if the appropriate transcriptomic data existed to support small

molecule upregulation of such a candidate microRNA. We thus began by working to determine if any natural microRNAs functioned to inhibit *DUX4*.

Prior gene expression studies reported microRNAs that were differentially expressed in FSHD cells compared to controls, but the biological significance of any of the putative FSHD-related miRNAs remained unclear and no *DUX4*-targeting miRNAs were identified in any of these studies[48–53]. In our study, differential gene expression in FSHD was not part of our screening criteria when identifying miRNA candidates. Indeed, we only considered one question to determine if a specific miRNA was potentially important—does it target *DUX4*? We selected 24 putative *DUX4*-targeting miRNAs from two different bioinformatics algorithms for empirical testing. This represented 1.3% of the 1917 known human miRNAs. We then used a battery of empirical tests to confirm—or mostly, refute—the ability of these 24 predicted miRNA candidates to silence *DUX4*. *MiR-675* was the only one that regulated *DUX4*, but did so powerfully and consistently using numerous different assays, and having multiple binding sites in both the ORF and 3′ UTR of the *DUX4* mRNA. This single miRNA represents 0.05% of all known human miRNAs, and it is possible that *miR-675* is not the only *DUX4*-targeting miRNA—just the first identified. Additional work is required to mine the miRNome for additional *DUX4*-targeting microRNAs.

In addition to the RNAi mechanism we explored here, non-sense mediated decay (NMD) has also been shown to regulate *DUX4* mRNA degradation[54]. This mechanism involves a double-negative feedback loop involving NMD and *DUX4*. In this model, NMD normally degrades *DUX4* mRNA, but rare transcripts can escape NMD and become translated into DUX4 protein, which causes proteolytic degradation of UPF1, a key component of the NMD process. Thus, DUX4 protein inhibits NMD, thereby allowing local amplification of *DUX4* gene expression until enough DUX4 protein is produced to presumably kill the cell. Since translation and RNAi both occur in the cytoplasm, it is possible that *miR-675* and the miRNA pathway could target and reduce *DUX4* transcripts that escape NMD (Supplementary Fig. 20). This has implications for emerging gene silencing therapies for FSHD, which could help to avoid "bursts" of *DUX4* expression that escape NMD inhibition.

Having identified *miR-675* as a *DUX4* regulator, we then worked to leverage this finding for therapeutic purposes in FSHD. We first took a gene therapy approach to express *miR-675* in *DUX4*-expressing mouse muscle, using a strategy similar to the one we developed using artificial microRNAs (Fig. 8)[21,29,33,55]. Because *miR-675* was almost as potent an RNAi trigger as our artificial microRNA *miDUX4.405*, we were not surprised to find that over-expression of AAV-delivered *miR-675* prevented DUX4-associated toxicity and suppressed DUX4-responsive biomarkers in mouse muscle. Thus, our data support another avenue for RNAi-based gene therapy of FSHD, using a natural microRNA. Having established proof-of-principle for using *miR-675* as a potential therapy for FSHD, we next assessed the possibility of using small molecules to upregulate *miR-675* as potential *DUX4*-targeting therapeutics, which was the main aim of our study. Prior studies showed that *miR-675* was induced with treatment of melatonin, and estrogen alone or in combination with progesterone[43–45]. In this work, we tested melatonin, estrogen (β-estradiol), and progesterone (medroxyprogesterone acetate) analogs on HEK293 cells that usually express very minimal levels of *miR-675*, and on three FSHD muscle cell lines expressing low (15A), medium (18A) and high (17A) levels of *DUX4*. We confirmed that each of the compounds indeed increased *miR-675* in our cell culture systems and operated to decrease *DUX4* and the *DUX4*-responsive biomarker TRIM43

expression in all cell lines tested (Fig. 9a, b). Importantly, inhibiting *miR-675* with the anti-miR-675 antagomir or preventing its binding to a *miR-675*-resistant *DUX4* construct (CMV.DUX4 miR-675Res) supported that the three molecules exert their *DUX4* inhibitory effect through *miR-675* action (Supplementary Fig. 18). Additional experiments, whether by testing the 3 molecules on FSHD affected myotubes overexpressing *miR-675* (Fig. 9c), on healthy control myotubes (Supplementary Fig. 19a) or by assessing their effect on differentiation Supplementary Fig. S19b), consolidated this proposition. Interestingly, recent evidence showed that *miR-675* levels decrease in muscle with age and since FSHD is a progressive disease that tends to worsen with age, it is possible that *miR-675* reductions could mechanistically contribute to progression by having reduced inhibitory effects on *DUX4*[56,57,1–4]. Thus, by re-establishing high levels of *miR-675* in aging skeletal muscles, these small molecules could reduce DUX4-induced muscle damage.

Interestingly, estrogen or its derivative β-estradiol have been previously linked to FSHD pathogenesis, although the role of FSHD in the disease is not definitive. Some previous reports suggested that estrogen might be protective in FSHD disease, as females were reported to be less severely affected than males and may have persistent worsening of symptoms after childbirth and menopause[58–60]. One study suggested that the beneficial effects of estrogen were mediated by the estrogen receptor (ERβ), which, when activated by estrogen, sequesters the DUX4 protein in the cytoplasm of cells and prevents its toxic effects in nuclei[61]. In contrast to these reports suggesting a protective effect of estrogen, a recent study of FLExDUX4 mice reported that females perform worse than males in some outcome measures[62]. However, we note that ERT2 system used to generate these animals utilizes the anti-estrogen drug Tamoxifen to induce high levels of *DUX4* expression, thereby complicating interpretation of the impacts of estrogen on phenotypes in these animals. In addition, a recent clinical study of 85 female FSHD patients did not find a significant correlation between differences in estrogen exposure and disease severity[63]. Interestingly their clinical approach consisted of subtracting periods with high progesterone levels from the reproductive life span so a protective effect caused by interplay with other reproductive hormones, including progesterone, could not be ruled out. Here, our data suggest a mechanism by which estrogen, and/or estrogen and progesterone, could at least partially protect cells from FSHD disease by counteracting *DUX4* expression via *miR-675* upregulation.

Melatonin has been previously identified as a promising drug therapy for neuromuscular diseases due to its anti-inflammatory and antioxidant properties. For this purpose, it was tested in the *mdx5Cv* Duchenne muscular dystrophy (DMD) mouse model, where it improved muscle function and enhanced the redox status of the muscle[64]. In another study, melatonin prevented the premature senescence of cardiac progenitor cells that occurs in heart diseases. The study revealed that melatonin exerts its effect via the H19/*miR-675*/USP10 pathway in which *miR-675* targets the USP10 peptidase, leading to a downregulation of p53 and p21 proteins[43]. Some previous studies suggested a role of p53 in DUX4 toxicity, although this finding has not been universally supported in all systems[5,13,19,65–68]. Nevertheless, it is possible that in addition to targeting *DUX4*, *miR-675* could also target the ubiquitous USP10 protein in FSHD skeletal muscles, leading to a better outcome.

Finally, *miR-675* has been shown to possess the property of enhancing skeletal muscle regeneration and differentiation by targeting and down-regulating the anti-differentiation SMAD transcription factors (SMAD1 and SMAD5) critical for the bone morphogenetic protein (BMP) pathway and the DNA replication

initiation factor CDC6 (ref. [27] and Supplementary Fig. 2, 11b and 12). Alongside the silencing of *DUX4* expression, this property is expected to be beneficial for FSHD affected skeletal muscles as it could help regenerate new muscle fibers in which *DUX4* expression is then reduced. So far, the spatio-temporal conditions in which *miR-675* initiates regeneration and differentiation are still unknown. It is also unknown whether *miR-675* is required to be delivered to satellite cells to exert its regenerative properties. What we know is that *H19* accumulates and *miR-675* expression is triggered upon muscle injury[27,69]. In future studies, to account for the possible necessity to deliver *miR-675* into muscle fibers and satellite cells, AAV9 vectors could be used since they were shown to transduce satellite cells with better efficiency than other serotypes[70,71].

In summary, we identified *miR-675* as a natural DUX4 inhibitor that may have therapeutic applications for treating FSHD, and natural *miR-675* variations among individual people could influence *DUX4* expression, thereby justifying future examination of *miR-675* as a potential disease modifier. To our knowledge, this is the first study utilizing small molecule upregulation of a natural microRNA for therapeutic purposes. As such, our study provides support to more broadly characterize the DUX4-targeting miR-Nome; by identifying more DUX4 inhibitors, it may be possible to identify numerous drugs that could be used to upregulate other natural, DUX4-targeting microRNAs. Finally, this strategy can theoretically be employed to target any dominant disease gene using drug-based upregulation of natural microRNAs, and may therefore have broader implications for the RNAi therapy field.

## Methods

**Sequence generation and cloning**. The *miR-675* sequence was obtained from the miRBase database (www.mirbase.org). All *miR-675* constructs (*miR-675, miR-675-5p,* and *miR-675-3p*) were cloned into an expression plasmid downstream of an RNA polymerase III class of promoters (U6 or H1 promoters). The sequence of the stem loop structure of *miR-675, miR-675-5p,* and *miR-675-3p* constructs is shown in Supplementary Fig. 3. The CMV.H19 construct was purchased from OriGene. The non-toxic *DUX4.HOX1* DNA binding domain mutant was cloned as previously described[13]. The CMV.DUX4-FL/CMV.eGFP construct expressing the full length DUX4 (*DUX4-FL: DUX4 ORF* with the *3'UTR*) was PCR amplified and cloned into the AAV6.CMV pro-viral backbone plasmid as previously described[13]. The original V5 peptide tag was mutated to prevent mis-splicing of the V5 and DUX4 stop codon via recombinant PCR as previously described[42]. In addition, this plasmid contains a cytomegalovirus promoter (CMVp)-driven *DUX4* full-length sequence encompassing the *DUX4* open reading frame (*DUX4 ORF*), the *DUX4* 3'UTR sequence (pLAM sequence). The latter is formed of exon 1 (Ex1), exon 2 (Ex2), and exon 3 (Ex3) and the endogenous *DUX4* unconventional polyA sequence (ATTAAAA) (epA). The CMV.DUX4 miR-675Res construct expressing the *DUX4 ORF* mutant resistant to *miR-675* inhibition was cloned by recombinant PCR using EcoRI/KpnI restriction sites, using wild-type the DUX4 as template, and the following primers: forward: 5′ CCGAGAATTCCTCGACTTATTAATAGTA ATCAATTACGGGGTCA 3′, forward middle: 5′ ACCCAAGATCTGGGGCAA GGTGGGCAAAAGCCGGGAGGA 3′, reverse middle: 5′ CACCTTGCCCCAGA TCTTGGGTGCCTGAGGGTGGGAGAG 3′, reverse: 5′ CGGG TACCCTACGT AGAATCGAGCCCGAGGAG 3′. The CMV.DUX4-miR-675Res contains a CMVp-driven *DUX4 ORF* with silent point mutations in the high affinity, ORF-located *miR-675* binding site (see TS780M vs TS780WT sequence alignment), and has no *DUX4* 3'UTR. The absence of the latter eliminated the other *miR-675* target sites on the *DUX4* transcript. This construct also contains the mutated V5 epitope sequence (V5.2) and the SV40 polyadenylation signal (SV40 pA). To clone CMV.eGFP into the CMV.DUX4-FL and CMV.DUX4 miR-675Res plasmids, CMV.eGFP was amplified by PCR using CMV.eGFP as template and the following primers: forward: TTACTAGTATTAATAGTAATCAATTACGG, reverse: CAATGA ATTCGTTAATGATTAACCCGCCAT. The PCR product was then cloned in the plasmid backbone using SpeI/EcoRI restriction sites. To make RenLuc-PTS (reverse complement of every mature miRNA sequence used in Supplementary Fig. 1), we ordered an oligonucleotide containing target sites of every miRBase-predicted miRNA used in this study, and used recombinant PCR to fuse it as the 3'UTR of *Renilla* luciferase in the psiCheck2 (RenLuc) dual luciferase plasmid (Promega). A similar strategy was used to clone the perfect target site for *miR-675* at the 3'UTR of *Renilla* luciferase in RenLuc-miR-675R. To make the dual luciferase plasmids in which the *DUX4 ORF, DUX4 3'UTR,* or full-length *DUX4* (*DUX4-FL*) were inserted as the Renilla luciferase 3'UTR (RenLuc-DUX4 ORF, RenLuc-DUX4 3'UTR, and DUX4-FL), we PCR amplified these sequences using the CMV.DUX4-FLΔV5 plasmid as template, and cloned into the RenLuc. To make RenLuc-DUX4-FL expression plasmid (Fig. 1a), *DUX4-FL* (*DUX4 ORF* without V5 tag + *3'UTR*) was PCR amplified using CMV.DUX4-FLΔV5 as template with the following primers: forward: 5′ CCGGCTCGAGATGGCCCTCC CGACAC 3′, reverse: 5′ ACGAC TAGTGGGGAGGGGGCATTTTAATATAT CTC 3′. The PCR product was then cloned into a previously designed RenLuc SD5 mutant plasmid using XhoI/SpeI restriction sites and the RenLuc.SD5 mutant-DUX4 3'UTR plasmid backbone. The *Renilla* luciferase gene has a splicing donor mutation (*SD5) that prevents the alternative splicing of the *DUX4-FL* mRNA[42]. To make the RenLuc-DUX4 ORF-miR-675Res expression plasmid, we performed a recombinant PCR to delete one of the strongest *miR-675* target sites (TS780) in *DUX4 ORF* and eliminated the *DUX4* 3'UTR using the following primers: forward: 5′ CCGGCTCGAGATGGCCCTCCCCGACAC 3′, forward middle: 5′ CGGGCAAAAGCCGGGAGGA 3′, reverse middle: 5′ TCCTCCCGGCTTTTGCC CGGCCTGAGGGTGGGAGA 3′, and reverse: 5′ AGCGGCCGCAAGCTCCTCC AGCAGAGC 3′. This construct was then cloned into the RenLuc-backbone using XhoI/NotI restriction sites. A full list of primers is found in Supplementary Table 3.

**HEK293 cell culture**. HEK293 cells were obtained from ATCC [HEK-293 (ATCC® CRL-1573™)]. HEK293 cells were grown using DMEM (Gibco) medium supplemented with 20% FBS (Corning), 1% L-glutamine (Gibco) and 1% penicillin–streptomycin (Gibco). Transfected cells were grown in the same DMEM medium but lacking penicillin–streptomycin.

**Human myoblasts culture**. The following human muscle cell lines were used in this study: 15A FSHD human myoblasts (from a Male proband with 66 years of age), 17A FSHD human myoblasts (from a Male proband with 23 years of age), 18A FSHD human myoblasts (from a Female proband with 36 years of age) and 15V control human myoblasts (sister of 15A with 60 years of age). These cell lines were provided by the UMMS Wellstone Center biobank and have been previously characterized[39]. Previously immortalized 15A cell lines have a single 4qA permissive allele. Using immunocytochemistry (ICC), Jones et al. showed that 15A cell lines have 1:10$^4$ DUX4$^+$ nuclei, which is low when compared to other FSHD affected cell lines (i.e., 17A and 18A)[39]. 15V cell lines have two 4qB non-permissive alleles. We propagated cells by feeding them every two days with new LHCN medium [4:1 DMEM:Medium 199 (Gibco) supplemented with 15% characterized FBS (Corning), 0.02 M HEPES (Thermo Fisher), 0.03 μg/mL ZnSO$_4$ (Honeywell Fluka), 1.4 μg/mL Vitamin B12 (Sigma-Aldrich), 0.055 μg/mL dexamethasone (Sigma-Aldrich), 1% antibiotics/antimycotics (Gibco), 2.5 ng/mL hepatocyte growth factor (Millipore) and 10 ng/mL basic fibroblast growth factor (Millipore)]. For differentiation, we switched myoblasts to a differentiation medium [4:1 DMEM: Medium 199 (Gibco) supplemented with 15% KnockOut Serum Replacement (Thermo Fisher), 2 mM L-glutamine (Gibco), 1% antibiotics/antimyocytics (Gibco), 1 mM sodium pyruvate (Gibco) and 20 mM HEPES (Thermo Fisher)] when cells are at >90% confluency. Before adding differentiation medium, we washed cells with PBS (Gibco). We seeded cells with new differentiation medium every three days for up to 7 days. To detach cells, we used TrypLE express, phenol red (Life Technologies).

**Drug treatment**. HEK293 cells or human muscle cell lines were treated with β-estradiol (Sigma) or medroxyprogesterone acetate (MPA) (Acros organics) or melatonin (MP biomedicals) at various concentrations. We dissolved all three drugs in 100% ethanol that was used as a negative control (vehicle) treatment. We added all three drugs to HEK293 cells upon transfection. We added all three drugs to human muscle cells (15A, 17A, and 18A) at the 4th day of differentiation, 24 h prior to cell harvest.

**Dual-luciferase assay**. We performed this assay as previously described[29] and following the dual-luciferase reporter assay system (Promega) protocol, with some modifications. All plasmid constructs have the psiCheck2 dual luciferase reporter plasmid (Promega) as backbone that contains separate *Renilla* and *Firefly* luciferase genes, where the former contains the various target sequences used in this study, and the latter serves as a transfection normalizer. We cloned all *DUX4* and control sequences downstream of the *Renilla* luciferase stop codon, serving as a 3'UTR. We pre-plated HEK293 cells 24 h before transfection. We then co-transfected cells the luciferase DUX4 reporter and individual microRNA expression plasmids in an increasing luciferase DUX4 reporter:miRNA molar ratio using Lipofectamine 2000 (Invitrogen). The luciferase activity was measured 24 or 48 h after transfection. We determined DUX4 gene silencing as previously described[29].

**RNA extraction**. To perform Northern blot assay and QPCR on small RNA, we extracted total RNA from HEK293 cells and human myoblasts and myotubes using the miRVANA miRNA isolation kit (Thermo Fisher Scientific) following the manufacturer's instructions. To perform QPCR on genes other than *miR-675*, we used a Trizol (Ambion)-based RNA extraction protocol following the manufacturer's instructions.

**TaqMan quantitative RT-PCR gene expression assay**. We used this assay to quantify, via RT-qPCR, the expression of *pri-mir-675, miR-675-5p, miR-675-3p,*

*SMAD1*, *SMAD2*, *CDC6*, and *DUX4*-responsive biomarkers (*KHDC1L* and *TRIM43*). First, we started by eliminating genomic DNA from the RNA preparations, and then we generated cDNA using the High-Capacity cDNA Reverse Transcription Kit (Applied Biosystems) following the manufacturer's instructions. Within the same cDNA reaction, we used 1× RT random hexamer primers to generate the cDNA for *RPL13A* used as a reference gene, for *DUX4*-responsive biomarker genes and for *pri-mir-675*. To generate the cDNA for *miR-675*, we used the RT *miR-675* reverse primer provided by Thermo Fisher. We then performed the TaqMan gene expression assay using the TaqMan Gene Expression Master mix and TaqMan probes purchased from Thermo Fisher Scientific. We mixed 1× of probe with 1× of the TaqMan Gene Expression Master mix (Thermo Fisher), and with 20 ng of cDNA for *miR-675*, *SMAD1*, *SMAD2*, and CDC6, and ≥50 ng of cDNA for *DUX4*-responsive biomarkers. The *miR-675*-specific primers and probes are designed to quantify only the *miR-675-5p* and *-3p* mature sequences (see TaqMan Gene Expression Master mix protocol).

**SYBR green-based quantitative RT-PCR gene expression assay**. This assay was used to quantify *DUX4* expression. We eliminated genomic DNA using the dsDNase provided by the Maxima H Minus First Strand cDNA Synthesis kit with dsDNase kit. This DNase is engineered to only digest double stranded DNA sequences and is therefore incapable of digesting single stranded cDNA sequences. Even though, the dsDNase is heat inactivated before starting the cDNA synthesis step. We generated cDNA for *DUX4* and *H19* by using the Maxima H Minus First Strand cDNA Synthesis kit following the manufacturer's instructions, with some modifications. We used 25 pmoles of the Oligo(dT)$_{18}$ primer for the reverse transcription step (see manufacturer's protocol for cDNA synthesis). We mixed 1× RT buffer with 1× Maxima H Minus enzyme mix and added 25 pmoles of Oligo(dT)$_{18}$ primer, 0.5 mM dNTP mix, and dsDNase treated RNA (2.5 μg) to the mix. We modified the incubation protocol in order to extend the extension step to 2 h at 65 °C. In the QPCR step, we used the SsoAdvanced Universal SYBR Green Supermix from Bio-Rad. To do this, we mixed 1× of SsoAdvanced SYBR Green Supermix with at 125 ng of cDNA, and with 1 μM of forward and reverse primers. The latter are the same that were used in ref. [72]. The thermal cycling protocol was done following the manufacturer's instructions with some modifications. The DNA polymerase activation and DNA denaturation step were done at 95 °C for 30 s. The amplification step of 39 cycles was done using a denaturation step at 95 °C for 10 s and an annealing/extension and plate reading step at 60 °C for 30 s. We finished the amplification step by a melt-curve analysis step (snap cool to 3 °C followed by an increment of 0.5 °C every 5 s until reaching 95 °C). The QPCR protocol for *H19* was generously provided by Dr. Alexandra Belayew. In the QPCR step, we used 64 ng of cDNA with 0.3 μM of *H19* forward primer: CCGGCCTTCCTGAACA and *H19* reverse primer: TTCCGATGGTGTCTTTGATGT. All qRT-PCR was done using the Bio Rad CFX96 Real-time system.

**Digital droplet PCR (ddPCR)**. We performed RNA extraction and *DUX4* cDNA synthesis as described for QPCR above. The ddPCR reaction mixture (20 μL) contained 1X ddPCR Evagreen Supermix (Bio-Rad), 1 μM of forward and reverse *DUX4* primers (same as the ones used for QPCR), and 1 μL of 10-fold diluted cDNA. We generated droplets using the Automatic Droplet generator QX200 AutoDG (Bio-Rad), and amplified the reactions in a C1000 Touch™ Thermal Cycler with 96-Deep Well Reaction Module (Bio-Rad). We set up the cycling conditions following the QX200 ddPCR Evagreen Supermix protocol. Based on the forward and reverse primers melting temperature (T$_m$), we kept the annealing temperature at 60 °C. We then read droplets using the QX200 droplet reader (Bio-Rad). Finally, we analyzed data using the QuantaSoft analysis software 1.0 (Bio-Rad). For quantification, each reaction of 20 μL should give at least 10,000 acceptable droplets. To quantify *hsa-miR-675-5p*, *hsa-DUX4*, *hsa-RPL13A*, *hsa-TRIM43*, *mmu-RPL13A*, *mmu-Wfdc3*, and *mmu-Trim36* using TaqMan probes, we used a ddPCR reaction mixture (20 μL) containing 1× of ddPCR Supermix for probes (No dUTP), 1× of a specific TaqMan probe (ThermoFisher) and 50 ng of synthesized cDNA. We set up the cycling conditions following the QX200 ddPCR Supermix for probes protocol with annealing temperature at 58 °C.

**Prediction of *miR-675* target sites**. We used the mirBase, PITA and RNAhybrid algorithms [23,28] (reference for miRBase) to predict *miR-675* target sites within the *DUX4* full length sequence [29].

**Molecular beacon-binding assay and prediction of *miR-675-5p* target sites**. We predicted *miR-675* binding sites on DUX4 using the combination of the PITA and RNAhybrid algorithms [23,28]. We performed the molecular beacon-binding assay as previously described in ref. [29], with some modifications. We designed the DNA *miR-675-5p* molecular beacon following the same design described in ref. [73]. We performed the binding assay using the CFX Connect Real Time system apparatus. After mixing the molecular beacon and target site, we initiated the binding with a denaturation step at 95 °C for 3 min, followed by an annealing step at 45 °C for 10 min (Supplementary Data 1 and Supplementary Fig. 3). We measured fluorescence at the end of the annealing step. We incubated 200 nM of the mature *miR-675* DNA molecular beacon with target sites at concentrations ranging from 0.1 μM to

80 μM. We represented our data as Relative Fluorescent Units (RFU) (subtracted from background fluorescent signal) and determined the binding affinity ($K_d$) by fitting RFU values to a one site-specific binding Hill slope equation.

**Small-transcript northern blots**. We performed this blotting as previously described with some modifications [74]. We transfected *miR-675* and *H19* expression plasmids into HEK293 cells using Lipofectamine 2000 (Thermo Fisher). We then harvested and lysed cells 48 h after transfection using the miRVANA miRNA isolation kit following the manufacturer's instructions (Thermo Fisher Scientific). We did not enrich for miRNAs. We loaded and electrophoresed 50 μg of purified and DNase digested RNA on a 15% acrylamide-bisacrylamide (19:1) gel containing 8 M Urea (48%, wt/vol) and 1× Tris-Borate-EDTA (TBE) RNase free solution (Invitrogen). As a size reference, we used the miRNA marker from New England BioLabs (NEB) as miRNA ladder at 1:10 dilution. We designed and used a dual labeled (with two biotin tags at the 5′ and 3′ ends) DNA oligonucleotide probe specific to the *miR-675* guide strand (*miR-675-5p*) at 0.3 pmol (*miR-675-5p* probe: 5′biotin-CACTGTGGGCCCTCTCCGCACCA-3′biotin). Finally, we developed the blot using the Chemiluminescent Nucleic Acid Detection Module Kit from Thermo Fisher following the manufacturer's instructions and Hyblot CL autoradiography film optimized for chemiluminescence.

**Western blots**. Western blot to detect DUX4 protein. To assess the inhibition of DUX4 protein expression by *miR-675* and *H19*, we co-transfected CMV.DUX4-FL/ CMV.eGFP or CMV.DUX4-miR-675Res expression plasmids and either U6.MIR675, H1.MIR675, CMV.H19 or U6.miLacZ and *U6.miDUX4.405* controls to HEK293 cells using Lipofectamine 2000 (Thermo Fisher). We extracted total protein 48 h later with RIPA buffer containing 50 mM Tris (pH 7.5–8.0), 150 mM NaCl, 0.1% (v/v) SDS, 0.5% (v/v) deoxycholate, 1% (v/v) triton X-100 (Fisher Scientific) and 1 tablet of protease inhibitor (Thermo Fischer) per 10 mL of buffer. We quantified total protein extract using the DC Protein Assay (Bio-Rad). We then separated twenty microgram samples on 12% SDS-PAGE and transferred them to PVDF membrane using the wet transfer system. We then incubated PVDF membranes with the following antibodies for overnight at 4 °C: mouse monoclonal antibody to V5 (horseradish peroxidase [HRP]-coupled) [1:5000 in 5% milk TBST buffer (150 mM NaCl, 50 mM Tris-HCl pH 7.4 and 0.1% Tween 20 (Fisher Scientific)), R961-25; Invitrogen]; rabbit polyclonal eGFP antibody (1:50,000 in 3% BSA PBS, ab290; Abcam). Finally, we washed V5-probed blots five times, 5 min each with TBST buffer supplemented with 0.1% Tween 20. When using the eGFP antibody, we blocked PVDF membranes 30 min with 3% BSA PBS prior to blotting. We then washed eGFP-probed blots five times, 5 mins each with TBST buffer supplemented with 0.5% Tween 20 (Fisher Scientific). We incubated the washed PVDF membrane with HRP-coupled goat anti-rabbit secondary antibody (1:250,000 in 3% BSA PBS, 115-035-144; Jackson ImmunoResearch) for 1–2 h at room temperature. When we performed blinded western blots, we asked a lab member who is not involved in the study to blind the DUX4 and miRNA expression plasmids before we transfect them into HEK293 cells. We also performed the SDS-PAGE gel and immunoblotting under blinding conditions. At the end of every experiment, the same lab member unblinded the blots.

To assess the inhibition of Cdc6 protein expression by *miR-675* and *H19*, we transfected 15A FSHD myoblasts with either *miR-675* or *H19* expression plasmids and differentiated them for 7 days. We detected Cdc6 using the Cdc6 rabbit monoclonal antibody (C42F7; Cell signaling technology) at 1:500 dilution in 5% BSA TBST buffer. We washed Cdc6-probed blots 3 times, 10 min each with TBST buffer supplemented with 0.1% Tween 20 (Fisher Scientific), and then incubated them with HRP-coupled goat anti-rabbit secondary antibody (1:100,000 in 5% BSA TBST buffer) for 2 h at room temperature.

We detected α-tubulin using the α-tubulin rabbit polyclonal antibody (1:500 in 5% milk TBST buffer, ab15246; Abcam). We washed α-tubulin-probed blots five times, 5 mins each with TBST buffer supplemented with 0.1% Tween 20 (Fisher Scientific), and then incubated them with HRP-coupled goat anti-rabbit secondary antibody (1:100,000 in 5% milk TBST buffer) for 2 h at room temperature. We developed all blots by exposing them to X-ray films following treatment with Immobilon Western HRP substrate (Millipore) (Supplementary Figs. 11–14).

*β-actin* was detected using a monoclonal antibody produced in mouse (1:1000 in 5% milk TBST buffer, SIGMA) (Fig. 7a and Supplementary Fig. 13). We quantified protein levels using ImageJ.

**Apoptosis assay**. To measure Caspase 3/7 activity in HEK293 cells, we co-transfected these cells with *miR-675* or *H19* expression plasmids and CMV.DUX4-FL WT expression plasmid (not encompassing CMV.eGFP) using Lipofectamine 2000. 48 h after transfection, we treated cells with the Apo-ONE® Homogeneous Caspase-3/7 Assay (Promega) following the manufacturer's instructions. We measured the Caspase 3/7 activity (RFU) using a fluorescence microplate reader.

To measure Caspase 3/7 activity in human myoblasts, we electroporated *miR-675*, *H19* expression plasmids or *miR-675* antagomir (anti-miR-675, Thermo Fisher) into these cells using the high efficiency electroporation protocol, and were let to recover for 24 h in their growth medium (LHCN medium) before starting differentiation using the KOSR (induces *DUX4* expression [75]) supplemented

differentiation medium. We then let cells to differentiate for 7 days before reading the Caspase 3/7 activity.

**Alkaline comet assay**. We co-transfected HEK293 cells as in the apoptosis assay, and collected them 24 or 48 h after transfection. We performed the alkaline comet assay as previously described in Wang et al[36]. and Dmitriev et al[37]. without formamidopyrimidine DNA-glycosylase (FPG) enzyme treatment. For quantification of DNA damage, we used Tritek CometScore software 2.0 to analyze 10 images of randomly selected non-overlapping cells. To evaluate the extent of DNA damage, we measured the comet tail moment (the measure of tail length multiplied by tail intensity) for each nucleus, and represented values as the average comet tail moments of 30 to 60 nuclei.

**Flow cytometry**. For viability, we suspended cells in PBS containing 1:100 dilution of LIVE/DEAD® Fixable Near-IR stain. We then incubated cells for 30 minutes at room temperature and fixed them with 1% formaldehyde for 10 minutes at room temperature. We then washed the cells one time and re-suspended them in FACS buffer (2%FBS with 0.1% sodium azide in PBS) before flow cytometry analysis. We gated viable cells by staining them with the LIVE/DEAD® Fixable Near-IR stain at the 633 nm excitation wavelength. We gated viable cells expressing GFP fluorescent signal using the 488 nm excitation wavelength. We analyzed all samples using the Behemoth BD LSR II Flow Cytometer from BD Biosciences. We finally calculated percent of GFP-positive cells based on the number of total live cells using the flow cytometer software FlowJo 10.7.1.

**Mice**. In this study, we used 6- to 9-week-old C57BL/6 male mice. Mice were housed in the vivarium of Nationwide Children's Hospital. All housing rooms are set to a 12:12 light/dark cycle. Temperature and humidity are controlled by Johnson Controls. Rodent rooms are set to a temperature of 72 degrees. Humidity is maintained at 30–70%. For housing, we use Lab Products Super-mouse caging (Envirogard or RAIR) with automatic reverse osmosis (RO) watering. All cages are washed using 180-degree water and STERIS Cage-Klenz 180 and 200 detergents. This study complied with all relevant ethical regulations for animal testing and research. Animal studies were performed following the NIH guide for the care and use of laboratory animals. Animal studies received ethical approval and were supervised by the Abigail Wexner Research Institute at Nationwide Children's Hospital (NCH) Institutional Animal Care and Use Committee (IACUC). This study was performed under the NCH IACUC approval number: AR13-00015.

**AAV Vector delivery to mice**. For the bioactivity, 6- to 9-week-old C57BL/6 male mice received direct 40 µL i.m. injections into the TA. Mice received adeno-associated virus scAAV6.CMV.DUX4-FL at $3 \times 10^9$ DNase-resistant particles (DRP) co-injected with scAAV6.milacZ at $5 \times 10^9$ DRP, and a contralateral co-injection of scAAV6.CMV.DUX4-FL at $3 \times 10^9$ DRP and scAAV6.U6.MIR675 at $5 \times 10^{10}$ DRP. Muscles were harvested 2 weeks post-injection. We injected scAAV6.H1.MIR675 at $5 \times 10^{10}$ DRP into the TA of C57BL/6 male mice. We injected saline control into the contralateral TA. All animal studies were performed according to the NIH Guide for the Care and Use of Laboratory Animals.

**Histology**. Dissected TA muscles were placed in O.C.T. Compound (Tissue-Tek) and frozen on liquid nitrogen-cooled isopentane. Cryosections were cut at 10 mm and then stained with H&E following standard protocols[76]. Images of Skeletal muscle sections were taken using the Olympus BX61 microscope and the Olympus cellSens standard 1.9 software.

**In situ Immunofluorescence**. Subcellular localization of DUX4 protein in cells was visualized using 1:100 DUX4 antibody (mouse mAb 9A12 from Millipore Sigma) and following the previously published protocol[7]. On muscle tissue sections, DUX4 protein was visualized using V5 immunofluorescence as previously described in ref. [8]. Images of cell cultures and Skeletal muscle sections were taken using the Olympus BX61 microscope and the Olympus cellSens standard 1.9 software.

**Reporting summary**. Further information on research design is available in the Nature Research Reporting Summary linked to this article.

## Data availability

All data supporting the findings described in this manuscript are available in the article and in the Supplementary Information, and from the corresponding author upon reasonable request. Uncropped and unprocessed scans of all blots are supplied in the Source Data file and/or as a Supplementary figure in the Supplementary materials file. miRbase database can be found on www.miRbase.com. PITA target prediction algorithm can be found on https://genie.weizmann.ac.il/pubs/mir07/mir07_prediction.html. Source data are provided with this paper.

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

## Acknowledgements

We acknowledge the FSHD Society (FSHS-22016-5; FSHS-82017-1; FSHS-82018-3; all to Nizar Y. Saad) and Friends of FSH Research for funding this project. During the course of this project, Dr. Harper's salary was partially funded by NIH (NIAMS) R01 (1R01AR062123-01); The University of Massachusetts Wellstone Center 5 P50 HD060848-13; and Nationwide Children's Hospital NIH NIAMS P50 Center of Research Translation (CORT) P50AR070604.

## Author contributions

Conceptualization, N.Y.S., and S.Q.H.; Methodology, N.Y.S., R.L.B., and S.Q.H.; Validation, N.Y.S., M.A., S.E.G., A.P., M.A.H., and G.A.C.; Formal analysis, N.Y.S., M.A., and S.E.G.; Investigation, N.Y.S. M.A., S.E.G., A.P., M.A.H., and G.A.C.; Resources, N.Y.S. and S.Q.H.; Writing—original Draft, N.Y.S.; Writing—review & editing, N.Y.S., R.L.B., and S.Q.H.; Visualization, N.Y.S.; Supervision, N.Y.S. and S.Q.H.; Project administration, N.Y.S.; Funding acquisition, N.Y.S. and S.Q.H.

## Competing interests

N.Y.S. and S.Q.H. declare the following competing interests: provisional patent application N° 63/145,255, covering *miR-675* upregulation as a therapeutic strategy for FSHD. M.A., S.E.G., G.A.C., M.A.H., A.P., and R.L.B. declare no competing interests.
