## [Peer Review File · Nature Communications]

Reviewers' Comments:

Reviewer #1:

Remarks to the Author:

The study by Saad et al. discovers that mir-675 targets DUX4, nicely maps the activity to a specific high activity site in the DUX4 ORF, and develops a gene therapy approach to treat FSHD based on overexpression of mir-675. The idea is provocative and the identification of a miRNA targeting DUX4 is very interesting for the field, therefore the impact of the study is appropriate for Nature Communications.

Overall the study is well done, with appropriate replications and statistics, and appropriate interpretations. I do have several concerns that would be important to address:

Major concerns:

1. In Figure 4D, using GFP as a reporter gene, because fluctuations in GFP+ cell number can be due to transfection effects the most appropriate assay for activity is not %GFP+ cells, which is essentially a binary readout, but mean GFP fluorescence of the GFP+ population. Please provide an example FACS plot of the reporter-transfected cells with mir-675 or control, and provide the mean GFP fluorescence of the gated GFP+ population, to replace the current graph.

2. The inability to detect endogenous DUX4 and determine whether it is increased in primary FSHD myotubes (Figure 6) is a weakness of the study. While the authors attempt to detect DUX4 by western blot, most others have simply used immunostaining and counted DUX4+ nuclei. Without evaluating the effect of endogenous mir-675 on endogenous DUX4 in patient cells, the impact of the study is significantly attenuated. RTPCR for target genes mitigates this somewhat, but only 2 genes are evaluated and the effects are quite moderate. I would suggest either performing RNA-seq on FSHD patient samples treated with anti-mir-675 to evaluate all DUX4 target genes vis-à-vis the whole genome, or perform immunostaining for DUX4 and counting of number of DUX4+ nuclei in FSHD myotubes. The authors mention that it is difficult to detect, however many groups have done this, so with some optimization, it should be possible.

3. Appropriate controls are lacking for figure 8: effect of compounds on TRIM43 expression also needs to be evaluated in control myotubes. If they are acting through DUX4 (via mir-675), then there will be no effect in control myoblasts. This point should also be supported using immunostaining for nuclear DUX4.

4. Are estrogen, progesterone and melatonin inhibiting DUX4 through mir-675, or through some other mechanism? This should be tested directly by overexpressing mir-675 and seeing if the drugs maintain an effect. In the presence of saturating mir-675, the drugs should have no effect.

5. The result that mir-675 targeting DUX4 resides within the H19 gene is really interesting, given the effect of H19 imprinting on skeletal muscle development. The impact of the paper would probably be increased by mentioning H19 in the abstract.

Minor concerns:

The phrase "required introducing two new sets of reagents into our experimental toolbox" (line 359) is overly-colloquial. "required two new sets of reagents" would be better.

The three graphs in Figure S16 are unlabeled. I assume the left is mir-675, the middle is DUX4, and the right is TRIM43.

Positive comment: the schematics of the miRNAs are helpful (Fig. 1A and 2B).

Reviewer #2:

Remarks to the Author:

The authors use publicly available programs to computationally identify mir-675 as potentially targeting the DUX4 transcript and show that expression of mir-675-5p decreases expression of a reporter with DUX4 sequences. A molecular beacon binding assay confirmed binding of mir-675 to several of the predicted DUX4 sequences, with two sites showing the highest affinity. Co-transfection assays in 293 cells showed that mir-675 diminished DUX4 mRNA and protein dependent on these two strong binding sites, and rescued DUX4 cell toxicity by several assays (however, the constructs with weaker mir-675 expression (e.g., CMV-H19) and weaker DUX4 protein suppression showed equal or better rescue of toxicity, and this is shown later in FSHD cells in Suppl Fig 14B) and activation of a DUX4-regulated promoter. As previously shown, H19 increases during muscle cell differentiation and, as now shown, mir-675-5p shows a modest increase despite the robust H19 increase. An antagomir of mir-675 increased a reporter with DUX4 sequences, even with the two strongest binding sites mutated, and increased DUX4 regulated genes in FSHD muscle cells. The antagomir also might have increased transfected DUX4 protein level in FSHD cells (see point 4). Delivery of DUX4 to mouse muscles by AAV caused damage, central nuclei, and activation of DUX4 target genes that was largely prevented by co-administration of AAV-mir-675. Immunodetection also shows loss of the staining for the tagged DUX4, but the images appear to show cytoplasmic staining. Three molecules shown to increase mir-675 also showed mild to moderate increase in mir-675 in three different FSHD myotube cultures, which was associated with mild to moderate decrease in DUX4 and TRIM43 RNAs.

This is an important study with good support for most of the main conclusions. There are some aspects the authors might revise to better support some of their findings and to improve the presentation.

Major critiques:

1. The level of detail provided might be justified as presenting a complete record, but this reviewer would have preferred a more focused presentation of the experiments critical to support the conclusions. Further, the narrative style providing rationale, justification, and interpretation in the Results section was, again in this reviewer's opinion, distracting rather than informative.

2. Figure 5 shows a dramatic increase in H19 RNA with muscle differentiation, consistent with prior studies. However, mir-675-5p increases marginally or at most 2-fold depending on the cell line. How then to interpret the biological activity of the CMV-H19 or the role of endogenous H19. Some acknowledgement or discussion would help the reader.

3. Figure 6c shows increased DUX4 activity following antagomirs in FSHD muscle cultures, but the endogenous DUX4 is not measured (see next point as well). Because Figure 1c apparently shows decreased DUX4 containing RNA transcripts due to mir-675, it is not clear why the authors do not measure the endogenous DUX4 mRNA in addition to its targets in Figure 6c. Also, immunodetection of DUX4 protein could be used to determine the percentage of DUX4 positive nuclei, although I understand that the authors might not want to do this because quantification is difficult and I do not feel it is absolutely necessary.

4. In the text describing the results of Figure 6D, the authors state that "After four days of differentiation, we detected minimal amounts of the transfected DUX4-FL in one of the three replicates." Only after reviewing the Figure 6C and Suppl Fig S13 is it evident that no DUX4 was detected in the other replicates and this sentence should be edited for clarity. However, the major concern with this data is that cells expressing the DUX4-FL generally die in about 48 hrs, whereas in this experiment the transfection is done in myoblasts followed by 4 days in differentiation conditions. This might explain the near absence of DUX4, but also raises questions about the high levels of DUX4-mir-675Res expression and whether it has less toxicity. For this reason, I feel that this is a flawed experiment that needs additional studies or another approach to test the point. For example, looking at endogenous DUX4 mRNA as suggested in point 3.

5. The findings in Figure 7 are dramatic. It is surprising that this would be achieved with relatively modest decreases in DUX4 mRNA, indeed some muscles with mir-675 had as much DUX4 mRNA as the miLacZ muscles. While Trim38 shows suppression, Wfdc3 not so much. This makes measuring the DUX4 protein critical. The immunodetection in muscle sections is not convincing for

this. The staining appears cytoplasmic rather than nuclear, unless I am misreading the image as the DAPI is very hard to see. I would like to see whether a western shows a more robust decrease of DUX4 protein than is seen for the DUX4 RNA.

6. Figure 8 shows decreased DUX4 and its target genes following treatment with the selected drugs that have a modest induction of mir-675. Because the expression of DUX4 depends on the differentiation state of muscle, including some markers of muscle differentiation would be important to show that this is not due to altering differentiation by the drugs. Although outside the scope of what is fair to ask, a mir-675 knock out would greatly strengthen this experiment.

Minor points:

7. Line 56 states that DUX4 is normally expressed at the human 2-cell stage, whereas this should state at the 4-cell stage (in mouse Dux is expressed at the 2-cell stage).

8. Line 70 states that there are only two oligonucleotide-based RNAi drugs. Since many readers might not quickly distinguish RNAi mechanisms from other oligonucleotide-based mechanisms, it might be confusing without a specific definition of RNAi added here or elsewhere in the manuscript.

9. Line 75 states that gene therapy is currently the best approach for delivering RNAi to muscle for FSHD. This is a valid opinion, but since no gene therapy for skeletal muscle is approved, nor has definitively shown clinical effectiveness in trials, it might be more clear to either edit for accuracy or include that this is an opinion.

10. In Fig 3B the Wt sequence is incorrect.

RE: Point by point response to referees for revision of NCOMMS-20-26218A-Z

Dear referees,

Please find attached our revised manuscript entitled “Human miRNA *mir-675* inhibits *DUX4* expression and may be exploited as a potential treatment for Facioscapulohumeral muscular dystrophy”. We thank you for helpful comments and suggestions. As you know, we have all encountered a global pandemic this past year or more, which negatively impacted our ability to do research for many months. We therefore appreciate your patience as we worked to perform additional experiments to address the reviews of the first submission and improve the paper. We have now made substantial changes and updates to the manuscript in response to your comments and requests noted below.

The paper represents a substantial amount of work. Our original submission contained:

- 8 Figs, 2 Tables, 16 Supplemental Figures, and 5 Supplemental Tables
- The main manuscript file was 69 double-spaced pages containing 19,280 words with refs.

Reviewer 2 indicated that our paper provided too much detail, and we agreed that it needed to be edited for greater brevity. Nevertheless, we note that the paper was already dense with data, and in response to reviews, we performed new experiments and added several new pieces of data to the manuscript. In particular, in the main manuscript, Figures 4D, 5B-D, 6A-B and 7B are new. In addition, our new Supplementary Figures S13 and S14 contain raw western blots to support Figs 5C and 6A, and all of Supplementary Figure S16 is new, in response to a reviewer’s comment. Finally, in this revision, we also included 3 Supplementary Powerpoint files showing all stained muscle sections from in vivo experiments included in this manuscript. Thus, our revised manuscript now contains:

- 7 Figs, 2 Tables, 17 Supplemental Figures, and 5 Supplemental Tables, 3 Supplemental PPTs
- The main manuscript file is now 64 double-spaced pages containing 17,395 words with refs.

Thus, despite adding new data to 4 figures and 6 additional supplementary figures/files, the main manuscript is now 10% shorter than the original submission and we hope the readability is improved. All changes in the manuscript are colored in **blue** text. Our point-by-point responses to the reviews are included below.

Sincerely,

Scott Q. Harper, Ph.D.
Professor of Pediatrics, The Ohio State University and
Principal Investigator, Center for Gene Therapy,
The Research Institute at Nationwide Children’s Hospital
Columbus, OH 43205

Reviewer 1 remarks:

Comment 1: In Fig 4D, using GFP as a reporter gene, because fluctuations in GFP+ cell number can be due to transfection effects, the most appropriate assay for activity is not %GFP+ cells, which is essentially a binary readout, but mean GFP fluorescence of the GFP+ population. Please provide an example FACS plot of the reporter-transfected cells with mir-675 or control, and provide the mean GFP fluorescence of the gated GFP+ population, to replace the current graph.

Response: We thank you for this suggestion. We have now added these data to the paper in **Fig 4D**, and discuss the data in the Results section on Page 11, Lines 273-279.

Comment 2: The inability to detect endogenous DUX4 and determine whether it is increased in primary FSHD myotubes (Figure 6) is a weakness of the study. While the authors attempt to detect DUX4 by western blot, most others have simply used immunostaining and counted DUX4+ nuclei. Without evaluating the effect of endogenous mir-675 on endogenous DUX4 in patient cells, the impact of the study is significantly attenuated. RTPCR for target genes mitigates this somewhat, but only 2 genes are evaluated and the effects are quite moderate. I would suggest either performing RNA-seq on FSHD patient samples treated with anti-mir-675 to evaluate all DUX4 target genes vis-à-vis the whole genome, or perform immunostaining for DUX4 and counting of number of DUX4+ nuclei in FSHD myotubes. The authors mention that it is difficult to detect, however many groups have done this, so with some optimization, it should be possible.

Response: We performed the immunostaining experiment suggested by the reviewer and now added this data to the revised paper in **Fig 5D**. We also performed ddPCR to detect *DUX4* transcript, and these data are now included in **Fig 5B** of the revision. We discuss the data in the Results section on Page 13, Lines 315-320, and on Page 14, Lines 346-350.

Comment 3: Appropriate controls are lacking for figure 8: effect of compounds on TRIM43 expression also needs to be evaluated in control myotubes. If they are acting through DUX4 (via mir-675), then there will be no effect in control myoblasts. This point should also be supported using immunostaining for nuclear DUX4.

Response: We tested the effect of compounds on *TRIM43* expression in control myotubes (15V and 18U). Unfortunately, *TRIM43* was undetectable, even in vehicle-treated control cells. The absence of *TRIM43* is consistent with the notion that it is a DUX4-activated gene in muscle, and that without DUX4, *TRIM43* is not present. Nevertheless, once again, *mir-675* expression increased in 15V myotubes treated with the 3 drugs (**Supplementary Fig. S16A**). We discussed these data in Supplementary Results (Page 6, Lines 141-150).

We also performed the experiment suggested by Reviewer 1, in which we over-expressed *mir-675* in the presence and absence of drugs and assess *TRIM43* expression. Please see Response to **comment 4** for details.

Finally, as suggested by this Reviewer, we supported our data using immunostaining for nuclear DUX4 in FSHD affected 18A myotubes treated with the compounds (**Fig. 7B**). DUX4 expression was lower in drug-treated cells versus those treated with vehicle. We discuss these data in the Results section on Pages 17 and 18 of the revised manuscript, Lines 431-439.

Comment 4: Are estrogen, progesterone and melatonin inhibiting DUX4 through mir-675, or through some other mechanism? This should be tested directly by overexpressing mir-675 and seeing if the drugs maintain an effect. In the presence of saturating mir-675, the drugs should have no effect.

Response: We performed the experiment and now included these data as **Supplementary Figure S16B**. Reviewer 1 suggested we over-express *mir-675* in the presence and absence of the drugs that upregulated *mir-675*, and then assess the effect [on *TRIM43* expression]. The idea here was that, to quote Reviewer 1: "In the presence of saturating *mir-675*, the drugs should have no effect." To do this, we transfected 15A FSHD

muscle cells with *mir-675*, and then treated cells with ethanol vehicle or drugs. Our results indeed showed no significant additional suppression of *DUX4* and *TRIM43* in the presence of saturating *mir-675*. We noted this on page 17 of the manuscript (lines 426-430) and provided detailed data in the Supplement (**Fig S16B**). In another experiment, we reasoned that the most straightforward approach to test this idea was to treat 15V cells (which do not express *DUX4*) with the 3 drugs and then measure *TRIM43* levels to determine if the drugs directly impacted *TRIM43*. This experiment was also suggested by reviewer 1. Unfortunately, we did not detect *TRIM43* at all in unaffected myotubes. We noted this in Supplemental results. This observation is consistent with the notion that *TRIM43* requires *DUX4*-mediated activation in muscle cells. Finally, we noted in the Discussion that previous studies suggested another mechanism by which estrogen could exert protective effects on FSHD cells by sequestering and inhibiting *DUX4* protein. This discussion was included in the original paper and is now located on page 21 of the revised manuscript (Lines 530-548). It therefore is possible that estrogen treatment could potentially enable a two-pronged attack on *DUX4* (via protein sequestration and RNAi mediated by *mir-675* upregulation).

Comment 5: The result that *mir-675* targeting *DUX4* resides within the *H19* gene is really interesting, given the effect of *H19* imprinting on skeletal muscle development. The impact of the paper would probably be increased by mentioning *H19* in the abstract.

Response: Thank you for this suggestion. We have now mentioned *H19* in the abstract, as follows: “Here we report a new strategy to direct RNAi against *DUX4* using the natural microRNA *mir-675*, which is derived from the lncRNA *H19*”.

Comment 6: Minor concerns: the phrase “required introducing two new sets of reagents into our experimental toolbox” (line 359) is overly colloquial. “required two new sets of reagents” would be better.

Response: We have changed this phrasing as suggested, Page 12, Lines 286-287.

Comment 7: The three graphs in Figure S16 are unlabeled. I assume the left is *mir-675*, the middle is *DUX4*, and the right is *TRIM43*.

Response: We apologize for the oversight and have now labeled the graphs appropriately.

Reviewer 2 remarks:

Comment 1: The level of detail provided might be justified as presenting a complete record, but this reviewer would have preferred a more focused presentation of the experiments critical to support the conclusions. Further, the narrative style providing rationale, justification, and interpretation in the Results section was, again in this reviewer’s opinion, distracting rather than informative.

Response 1: First, many of the co-authors of this study also perform peer-review activities and can appreciate the amount of time it must have taken the Reviewers to digest the large amount of data we packed into this manuscript. Furthermore, we are sorry that the detailed writing style was an impediment to efficiently review the original submission. We appreciate the Reviewer’s perspective and have attempted to improve the readability of the manuscript. To do this, we further edited the manuscript and also moved some materials that were in the initial manuscript to the supplement. Despite our efforts to cut text, we also note that several new pieces of data were added in response to the reviewers’ comments, which required additional new text. Thus, we took the Reviewer’s comment seriously, but the manuscript is still dense with data and overall, we were only able to cut the body of our revised manuscript by about 10%.

Comment 2: Figure 5 shows a dramatic increase in *H19* RNA with muscle differentiation, consistent with prior studies. However, *mir-675-5p* increases marginally or at most 2-fold depending on the cell line. How then to interpret the biological activity of the CMV-*H19* or the role of endogenous *H19*. Some acknowledgement or discussion would help the reader.

Response: First, we point out that we now moved the original **Fig 5** data to **Supplementary Fig S10**. Second, the Reviewer is correct to point out that *H19* levels are empirically much higher than *mir-675* levels. Indeed, this is true in general. *H19* functions as both a long non-coding RNA and a small percentage of the *H19* transcript can also be processed into *mir-675-5p* and *-3p*. Previous studies suggested the processing of *H19* to *mir-675* is inefficient, and our data could reflect this inefficiency (Cai and Cullen, “The imprinted *H19* noncoding RNA is a primary microRNA precursor”, *RNA*, 2007, PMID: 17237358.) To address the Reviewer’s comment, we have now added the following to the Supplement in the section where we discuss **Fig S10**: “*H19* expression was more pronounced than that of *mir-675* in myotubes. *H19* has other known functions as a full-length non-coding RNA beyond serving as a precursor of *mir-675*, and previous studies suggested processing of *H19* to *mir-675* is inefficient. The differences in expression level between *H19* and *mir-675* suggest the majority of *H19* remains unprocessed.”

Comment 3: Figure 6c shows increased DUX4 activity following antagomirs in FSHD muscle cultures, but the endogenous DUX4 is not measured (see next point as well). Because Figure 1c apparently shows decreased DUX4 containing RNA transcripts due to *mir-675*, it is not clear why the authors do not measure the endogenous DUX4 mRNA in addition to its targets in Figure 6c. Also, immunodetection of DUX4 protein could be used to determine the percentage of DUX4 positive nuclei, although I understand that the authors might not want to do this because quantification is difficult and I do not feel it is absolutely necessary.

Response: Reviewer 1 made a similar comment. We have now addressed this issue with the following:

- Assess endogenous DUX4 levels: we performed ddPCR and immunofluorescence staining to detect DUX4 in human FSHD myoblasts/myotubes. These data are now included in **Fig 5B** and **5D** and **Fig 7B**.
 - In Fig 5B, we used ddPCR to show increased *DUX4* mRNA levels following *mir-675* anti-*mir* treatment.
 - In Fig 5D, we stained 15A FSHD myotubes with the DUX4-specific mAB 9A12 antibody to show increased DUX4 protein expression following anti-*mir-675* or control treatment.
 - In Fig 7B, we stained 18A FSHD myotubes with the DUX4-specific mAB 9A12 antibody to show decreased DUX4 protein expression following treatment with the *mir-675* upregulating drugs β -estradiol, β -estradiol+MPA, and melatonin.

We discussed the data in the Results section on Page 13, Lines 315-320, and on Page 14, Lines 346-350.

Comment 4: In the text describing the results of Figure 6D, the authors state that “After four days of differentiation, we detected minimal amounts of the transfected DUX4-FL in one of the three replicates.” Only after reviewing the Figure 6C and Suppl Fig S13 is it evident that no DUX4 was detected in the other replicates and this sentence should be edited for clarity. However, the major concern with this data is that cells expressing the DUX4-FL generally die in about 48 hrs, whereas in this experiment the transfection is done in myoblasts followed by 4 days in differentiation conditions. This might explain the near absence of DUX4, but also raises questions about the high levels of DUX4-*mir-675*Res expression and whether it has less toxicity. For this reason, I feel that this is a flawed experiment that needs additional studies or another approach to test the point. For example, looking at endogenous DUX4 mRNA as suggested in point 3.

Response: We apologize for the confusing wording. We have now edited this section to clarify the point. It now reads: “After four days of differentiation, we detected minimal amounts of DUX4.V5 protein in one of the three replicates transfected with DUX4 alone (**Fig. 5C** and **Supplementary Fig. S13**). No DUX4 signal was found in the other two replicates.” Page 14, Lines 332-333.

In addition, we performed Caspase-3/7 assay to compare the death-inducing toxicity between the wild-type and *mir-675*-resistant *DUX4* constructs. This experiment supported that greater DUX4 expression causes greater cell death; thus *mir-675*-resistant *DUX4* does not have less toxicity. These data are now present in Figure 5C of the revision. Finally, we also note that we measured endogenous *DUX4* mRNA and protein using ddPCR and immunostaining as well (**Fig 5B-D**). We thank both Reviewers for these suggestions that served

to improve the paper.

We discussed these data on Pages 14-15, Lines 341-350, of the revised manuscript.

Comment 5: The findings in Figure 7 are dramatic. It is surprising that this would be achieved with relatively modest decreases in DUX4 mRNA, indeed some muscles with mir-675 had as much DUX4 mRNA as the miLacZ muscles. While Trim38 shows suppression, Wfdc3 not so much. This makes measuring the DUX4 protein critical. The immunodetection in muscle sections is not convincing for this. The staining appears cytoplasmic rather than nuclear, unless I am misreading the image as the DAPI is very hard to see. I would like to see whether a western shows a more robust decrease of DUX4 protein than is seen for the DUX4 RNA.

Response: Thank you for this suggestion. We have now performed western blots on extracts from injected mouse muscles. The original Figure 7 is now **Fig 6** in the revision, and the western data are now included in Fig 6, and raw western blot replicates are present in **Supplementary Fig S14**). These data show that V5-tagged DUX4 protein is lower in AAV.mir-675-treated animals compared to negative controls. We discuss these data on Page 16, Lines 388-391 of the revised manuscript.

Regarding immunodetection of DUX4 protein in muscle sections, we do indeed see cytoplasmic staining of DUX4 in some myofibers of this high-expressing DUX4 model, and have previously published this finding (21). We discuss these data on Page 16, Lines 396-398 of the revised manuscript.

Comment 6: Figure 8 shows decreased DUX4 and its target genes following treatment with the selected drugs that have a modest induction of mir-675. Because the expression of DUX4 depends on the differentiation state of muscle, including some markers of muscle differentiation would be important to show that this is not due to altering differentiation by the drugs. Although outside the scope of what is fair to ask, a mir-675 knock out would greatly strengthen this experiment.

Response: We thank the Reviewer for this observation and addressed this issue with the following:

- Assess the impact of the drugs on muscle differentiation. The concern by Reviewer 2 was that since *DUX4* is known to be activated by muscle differentiation, DUX4 levels would be reduced if the drugs inhibited differentiation. The Reviewer therefore suggested we determine the impact of the drugs on myoblast differentiation.
 - First, we argue that the timing of the experiment we initially performed already suggested that the drugs would not impact differentiation. Specifically, in studies using myoblasts, our approach involved differentiating cells for 4 days prior to adding drugs. Thus, the cells were already fated to myotubes prior to receiving the small molecules.
 - Nevertheless, we thought the reviewer made a solid point that *DUX4* levels increase during muscle cell differentiation and therefore performed the suggested experiment using 3 different FSHD myoblast cell lines: 15A, 17A, and 18A. Specifically, we differentiated cells for 4 days, then treated with drugs, and harvested RNA to measure two markers of muscle differentiation (*MYOG* and *MYH2*) using ddPCR. We found no significant impact of the 3 drugs on these differentiation markers compared to vehicle-treated cells. We now refer to these data on Page 17, Lines 426-430 of the revised manuscript, and detailed data are shown in **Fig S16C**.

Comment 7: Line 56 states that DUX4 is normally expressed at the human 2-cell stage, whereas this should state at the 4-cell stage (in mouse *Dux* is expressed at the 2-cell stage).

Response: Thank you for pointing out this inaccuracy. We changed the statement to 4-cell stage as suggested. Page 3, Line 49.

Comment 8: Line 70 states that there are only two oligonucleotide-based RNAi drugs. Since many readers might not quickly distinguish RNAi mechanisms from other oligonucleotide-based mechanisms, it might be confusing without a specific definition of RNAi added here or elsewhere in the manuscript.

Response: We have now added the following 3 sentences to the introduction on Page 3, Lines 57-62 of the revision: “RNAi is mediated by ~22 nucleotide (nt), non-coding microRNAs in complex with cellular gene silencing machinery, which together are called the RNA-induced silencing complex (RISC). The microRNA component of RISC base-pairs with target transcripts to trigger sequence-specific gene silencing. Importantly, artificial microRNAs can be engineered to trigger RNAi against disease genes.”

Comment 9: Line 75 states that gene therapy is currently the best approach for delivering RNAi to muscle for FSHD. This is a valid opinion, but since no gene therapy for skeletal muscle is approved, nor has definitively shown clinical effectiveness in trials, it might be more clear to either edit for accuracy or include that this is an opinion.

Response: We reduced the overstatement and added the following to the manuscript: “*Thus, gene therapy is currently the best approach for delivering an inhibitory RNA to muscle.*” This change is now on Page 4, Lines 71-72 of the revision.

Comment 10: In Fig 3B the Wt sequence is incorrect.

Response: Respectfully, we disagree with the Reviewer’s comment. Here we include the most recent Ensembl record for DUX4.

http://www.ensembl.org/Homo_sapiens/Gene/Summary?g=ENSG00000260596;r=4:190173774-190185942

The DUX4 cDNA is CCDS77990.

<https://www.ncbi.nlm.nih.gov/CCDS/CcidsBrowse.cgi?REQUEST=CCDS&DATA=CCDS77990>

Finally, here is the reference sequence on NCBI:

[https://www.ncbi.nlm.nih.gov/nucleotide/NG_034189.3?report=genbank&log\\$=nuclalign&blast_rank=1&RID=C2MA0W99013](https://www.ncbi.nlm.nih.gov/nucleotide/NG_034189.3?report=genbank&log$=nuclalign&blast_rank=1&RID=C2MA0W99013)

The WT DUX4 sequence shown in Fig 3B can be found in all these sequences. We have pasted in an image of CCDS77990 here with the WT sequence in Fig 3B highlighted.

M A L P T P S D S T L P A E A R G R G R R R R L V W T P S Q S E A L R A C F E R N P Y P G I A T
ATGGCCCT CCCGACAC CCTCGGAC AGCACCT CCCCGGG AAGCCCGG GGACGAGG ACGGCGAC GGAGACTC GTTTGGAC CCCGAGCC AAAGCGAG GCCCTGCG AGCCTGCT TTGAGCGG AACCCGTA CCCGGGA TCGCCACC

477

R E R L A Q A I G I P E P R V Q I W F Q N E R S R Q L R Q H R R E S R P W P G R R G P P E G R R
AGAGAACG GCTGGCCC AGGCCATC GGCATTCC GGAGCCCA GGGTCCAG ATTTGGTT TCAGAATG AGAGGTCA CGCCAGCT GAGGCAGC ACCGGCGG GAATCTCG GCCCTGGC CCGGGAGA CGCGGCC GCCAGAAG GCCGGCGA

288

K R T A V T G S Q T A L L L R A F E K D R F P G I A A R E E L A R E T G L P E S R I Q I W F Q N
AAGCGGAC CGCCGTCA CCGGATCC CAGACCGC CCTGCTCC TCCGAGCC TTTGAGAA GGATCGCT TTCCAGGC ATCGCCGC CCGGGAGG AGCTGGCC AGAGAGAC GGGCCTCC CGGAGTCC AGGATTCA GATCTGGT TTCAGAAT

432

R R A R H P G Q G G R A P A Q A G G L C S A A P G G G H P A P S W V A F A H T G A W G T G L P A
CGAAGGGC CAGGCACC CGGGACAG GGTGGCAG GGCAGCCG CGCAGGCA GGGGGCCT GTGCAGCG CGGCCCCC GGCGGGG TCACCTG CTCCCTCG TGGGTCGC CTTCGCC ACACCGGC GCGTGGG AACGGGG TTCCCGCA

576

P H V P C A P G A L P Q G A F V S Q A A R A A P A L Q P S Q A A P A E G I S Q P A P A R G D F A
CCCCACGT GCCCTGCG CGCCTGGG GCTCTCC ACAGGGG CTTTCGTG AGCCAGGC AGCGAGG CGGCCCCC GCGTGTCA GCCCAGCC AGGCCCGC CCGGCAGA GGGGATCT CCCAACCT GCCCGGC GCGCGGG ATTTCCGC

720

Y A A P A P P D G A L S H P Q A P R W P P H P G K S R E D R D P Q R D G L P G P C A V A Q P G P
TACGCGC CCCGCTC CTCCGAC GGGGGCT CTCCACC CTCAGGCT CCTCGGTG GCCTCCGC ACCCGGG AAAAGCCG GGAGGACC GGGACCCG CAGCGCGA CGGCCTGC CCGGCCCC TCGCGGT GGCACAGC CTGGGCC

864

A Q A G P Q G Q G V L A P P T S Q G S P W W G W G R G P Q V A G A A W E P Q A G A A P P P Q P A
GCTCAAGC GGGCCGC AGGCCAA GGGGTGCT TGCACCAC CCACGTCC CAGGGAG TCCGTGGT GGGGCTGG GGCAGGG TCACCAG TCGCCGG GCGGCTG GGAACCC AACCGGC GCAGCTCC ACCTCCC AGCCCGC

1008

P P D A S A S A R Q G Q M Q G I P A P S Q A L Q E P A P W S A L P C G L L L D E L L A S P E F L
CCCCCGA CGCCTCC CTTCCGC CGGCAGG GCAGATG AAGGCATC CCGCGCC CTCCAGG CGCTCCAG GAGCCGGC GCCCTGGT CTGCACT CCCTGCG CCTGTGC TGGATGAG CTCCTGGC GAGCCCG AGTTTCTG

1152

Q Q A Q P L L E T E A P G E L E A S E E A A S L E A P L S E E E Y R A L L E E L *
CAGCAGC GCAACCTC TCCTAGAA ACGGAGC CCCGGGG AGCTGGAG GCCTCGGA AGAGCCG CCTCGTG GAAGACC CCTCAGC AGGAAGAA TACCGGC TCTGTGG AGGAGCT TAG

Reviewers' Comments:

Reviewer #1:

Remarks to the Author:

The authors have done a significant amount of work to address the concerns, and their results support the conclusions of the study, so my recommendation is in favor of publication of the revised manuscript.

Reviewer #2:

Remarks to the Author:

The authors have addressed my prior concerns. I do have one new minor concern: The text does not make it clear whether the data in Figure 5B represents the expression and activity of the endogenous DUX4 or whether the cells were transfected with DUX4, as was done for the data in panels 5A and 5C. Please clarify the text.